# Predict, Refine, Synthesize: Self-Guiding Diffusion Models for Probabilistic Time Series Forecasting

**Marcel Kollovieh**[2][*][†]   **Abdul Fatir Ansari**[1][*]   **Michael Bohlke-Schneider**[1]

**Jasper Zschiegner**[1]   **Hao Wang**[1]   **Yuyang Wang**[1]

[1]AWS AI Labs   [2]Technical University of Munich

Correspondence to: `ansarnd@amazon.de`

## Abstract

Diffusion models have achieved state-of-the-art performance in generative modeling tasks across various domains. Prior works on time series diffusion models have primarily focused on developing conditional models tailored to specific forecasting or imputation tasks. In this work, we explore the potential of task-agnostic, unconditional diffusion models for several time series applications. We propose TSDiff, an unconditionally-trained diffusion model for time series. Our proposed self-guidance mechanism enables conditioning TSDiff for downstream tasks during inference, without requiring auxiliary networks or altering the training procedure. We demonstrate the effectiveness of our method on three different time series tasks: forecasting, refinement, and synthetic data generation. First, we show that TSDiff is competitive with several task-specific conditional forecasting methods (*predict*). Second, we leverage the learned implicit probability density of TSDiff to iteratively refine the predictions of base forecasters with reduced computational overhead over reverse diffusion (*refine*). Notably, the generative performance of the model remains intact — downstream forecasters trained on synthetic samples from TSDiff outperform forecasters that are trained on samples from other state-of-the-art generative time series models, occasionally even outperforming models trained on real data (*synthesize*).

## 1 Introduction

Time series forecasting informs key business decisions [27], for example in finance [47], renewable energy [56], and healthcare [7]. Recently, deep learning-based models have been successfully applied to the problem of time series forecasting [46, 31, 5]. Instantiations of these methods use, among other techniques, autoregressive modeling [46, 42], sequence-to-sequence modeling [54], and normalizing flows [43, 8]. These techniques view forecasting as the problem of conditional generative modeling: generate the future, conditioned on the past.

Diffusion models [48, 20] have shown outstanding performance on generative tasks across various domains [10, 28, 30, 36, 44] and have quickly become the framework of choice for generative modeling. Recent studies used conditional diffusion models for time series forecasting and imputation tasks [44, 52, 1, 6]. However, these models are task specific, i.e., their applicability is limited to the specific imputation or forecasting task they have been trained on. Consequently, they also forego the desirable *unconditional* generative capabilities of diffusion models.

---

[*]Equal contribution.
[†]Work conducted during an internship at Amazon.

37th Conference on Neural Information Processing Systems (NeurIPS 2023).

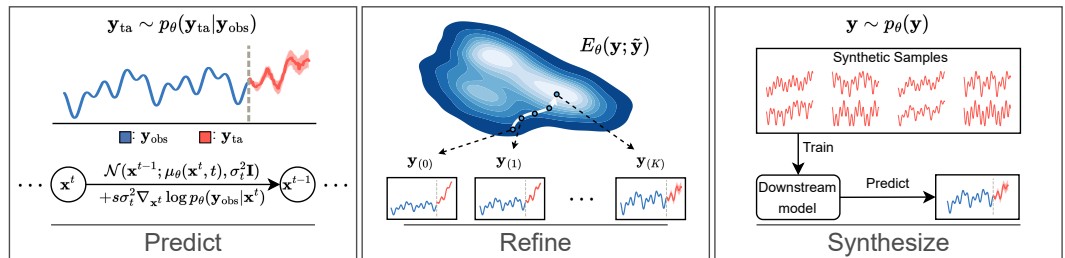

Figure 1: An overview of TSDiff's use cases. **Predict**: By utilizing observation self-guidance, TSDiff can be conditioned during inference to perform predictive tasks such as forecasting (see Sec. 3.1). **Refine**: Predictions of base forecasters can be improved by leveraging the implicit probability density of TSDiff (see Sec. 3.2). **Synthesize**: Realistic samples generated by TSDiff can be used to train downstream forecasters achieving good performance on real test data (see Sec. 4.3).

This raises a natural research question: *Can we address multiple (even conditional) downstream tasks with an unconditional diffusion model?* Specifically, we investigate the usability of task-agnostic unconditional diffusion models for forecasting tasks. We introduce TSDiff, an unconditional diffusion model for time series, and propose two inference schemes to utilize the model for forecasting. Building upon recent work on guided diffusion models [10, 19], we propose a self-guidance mechanism that enables conditioning the model during inference, without requiring auxiliary networks. This makes the unconditional model amenable to arbitrary forecasting (and imputation) tasks that are conditional in nature[3]. We conducted comprehensive experiments demonstrating that our self-guidance approach is competitive against task-specific models on several datasets and across multiple forecasting scenarios, without requiring conditional training. Additionally, we propose a method to iteratively refine predictions of base forecasters with reduced computational overhead compared to reverse diffusion by interpreting the implicit probability density learned by TSDiff as an energy-based prior. Finally, we show that the generative capabilities of TSDiff remain intact. We train multiple downstream forecasters on synthetic samples from TSDiff and show that forecasters trained on samples from TSDiff outperform those trained on samples from variational autoencoders [9] and generative adversarial networks [57], sometimes even outperforming models trained on real samples. To quantify the generative performance, we introduce the *Linear Predictive Score (LPS)* which we define as the test forecast performance of a linear ridge regression model trained on synthetic samples. TSDiff significantly outperforms competing generative models in terms of the LPS on several benchmark datasets. Fig. 1 highlights the three use cases of TSDiff: predict, refine, and synthesize.

In summary, our key contributions are:

- TSDiff, an unconditionally trained diffusion model for time series and a mechanism to condition TSDiff during inference for arbitrary forecasting tasks (observation self-guidance);
- An iterative scheme to refine predictions from base forecasters by leveraging the implicit probability density learned by TSDiff;
- Experiments on multiple benchmark datasets and forecasting scenarios demonstrating that observation self-guidance is competitive against task-specific conditional baselines;
- Linear Predictive Score, a metric to evaluate the predictive quality of synthetic samples, and experiments demonstrating that TSDiff generates realistic samples that outperform competing generative models in terms of their predictive quality.

The rest of the paper is organized as follows. Sec. 2 introduces the relevant background on denoising diffusion probabilistic models (DDPMs) and diffusion guidance. In Sec. 3, we present TSDiff, an unconditional diffusion model for time series, and propose two inference schemes to utilize the model for forecasting tasks. Sec. 5 discusses the related work on diffusion models for time series and diffusion guidance. Our empirical results are presented in Sec. 4. We conclude with a summary of our findings, the limitations of our proposals, and their potential resolutions in Sec. 6.

---

[3]Note that in contrast to meta-learning and foundation models literature, we train models per dataset and only vary the inference task, e.g., forecasting with different missing value scenarios.

## 2 Background

### 2.1 Denoising Diffusion Probabilistic Models

Diffusion models [48, 20] provide a framework for modeling the data generative process as a discrete-time diffusion process. They are latent variable models of the form $p_\theta(\mathbf{y}) = \int p_\theta(\mathbf{y}, \mathbf{x}^{1:T}) d\mathbf{x}^{1:T}$, where $\mathbf{y} \sim q(\mathbf{y})$ is the true underlying distribution. The latent variables $\{\mathbf{x}^1, \ldots, \mathbf{x}^T\}$ are generated by a fixed Markov process with Gaussian transitions, often referred to as the *forward process*,

$$q(\mathbf{x}^1, \ldots, \mathbf{x}^T | \mathbf{x}^0 = \mathbf{y}) = \prod_{t=1}^{T} q(\mathbf{x}^t | \mathbf{x}^{t-1}) \quad \text{and} \quad q(\mathbf{x}^t | \mathbf{x}^{t-1}) := \mathcal{N}(\sqrt{1 - \beta_t} \mathbf{x}_{t-1}, \beta_t \mathbf{I}), \quad (1)$$

where $\beta_t$ is the variance of the additive noise, $\mathbf{y}$ is the observed datapoint, and $q(\mathbf{x}^T) \approx \mathcal{N}(\mathbf{0}, \mathbf{I})$. The fixed Gaussian forward process allows direct sampling from $q(\mathbf{x}^t | \mathbf{y})$,

$$\mathbf{x}^t = \sqrt{\bar{\alpha}_t} \mathbf{y} + \sqrt{(1 - \bar{\alpha}_t)} \boldsymbol{\epsilon}, \quad (2)$$

where $\alpha_t = 1 - \beta_t$, $\bar{\alpha}_t = \prod_{i=1}^{t} \alpha_i$, and $\boldsymbol{\epsilon} \sim \mathcal{N}(\mathbf{0}, \mathbf{I})$. On the other hand, the reverse diffusion (generative) process is formulated as,

$$p_\theta(\mathbf{x}^0 = \mathbf{y}, \ldots, \mathbf{x}^T) = p(\mathbf{x}^T) \prod_{t=1}^{T} p_\theta(\mathbf{x}^{t-1} | \mathbf{x}^t) \quad \text{and} \quad p_\theta(\mathbf{x}^{t-1} | \mathbf{x}^t) := \mathcal{N}(\mu_\theta(\mathbf{x}^t, t), \sigma_t \mathbf{I}), \quad (3)$$

where $p(\mathbf{x}^T) \sim \mathcal{N}(\mathbf{0}, \mathbf{I})$ and $\sigma_t = \frac{1 - \bar{\alpha}_{t-1}}{1 - \bar{\alpha}_t} \beta_t$. The model is trained to approximate the true reverse process $q(\mathbf{x}^{t-1} | \mathbf{x}^t)$ by maximizing an approximation of the evidence lower bound (ELBO) of the log-likelihood. Specifically, $\mu_\theta$ is parameterized using a denoising network, $\boldsymbol{\epsilon}_\theta$,

$$\mu_\theta(\mathbf{x}^t, t) = \frac{1}{\sqrt{\alpha_t}} \left( \mathbf{x}^t - \frac{\beta_t}{\sqrt{1 - \bar{\alpha}_t}} \boldsymbol{\epsilon}_\theta(\mathbf{x}^t, t) \right), \quad (4)$$

which is trained to predict the sampled noise ($\boldsymbol{\epsilon}$ in Eq. 2) using the simplified objective function [20],

$$\mathbb{E}_{\mathbf{y}, \boldsymbol{\epsilon}, t} \left[ \| \boldsymbol{\epsilon}_\theta(\mathbf{x}^t, t) - \boldsymbol{\epsilon} \|^2 \right]. \quad (5)$$

By suitably adjusting the denoising neural network to incorporate the conditioning input, this objective can be employed to train both unconditional and conditional models.

### 2.2 Diffusion Guidance

Classifier guidance repurposes unconditionally-trained image diffusion models for class-conditional image generation [10]. The key idea constitutes decomposing the class-conditional score function using the Bayes rule,

$$\nabla_{\mathbf{x}^t} \log p(\mathbf{x}^t | c) = \nabla_{\mathbf{x}^t} \log p(\mathbf{x}^t) + \nabla_{\mathbf{x}^t} \log p(c | \mathbf{x}^t), \quad (6)$$

and employing an auxiliary classifier to estimate $\nabla_{\mathbf{x}^t} \log p(c | \mathbf{x}^t)$. Specifically, the following modified reverse diffusion process (Eq. 3) allows sampling from the class-conditional distribution,

$$p_\theta(\mathbf{x}^{t-1} | \mathbf{x}^t, c) = \mathcal{N}(\mathbf{x}^{t-1}; \mu_\theta(\mathbf{x}^t, t) + s \sigma_t^2 \nabla_{\mathbf{x}^t} \log p(c | \mathbf{x}^t), \sigma_t^2 \mathbf{I}), \quad (7)$$

where $s$ is a scale parameter controlling the strength of the guidance.

## 3 TSDiff: an Unconditional Diffusion Model for Time Series

In this section, we present our main contributions: TSDiff, an *unconditional* diffusion model designed for time series, along with two inference schemes that leverage the model for downstream forecasting tasks. We begin by outlining the problem setup and providing a concise overview of our network architecture. Subsequently, we delve into our first scheme — observation self-guidance — which enables conditioning reverse diffusion on arbitrary observed timesteps *during inference*. Secondly, we present a technique to iteratively refine predictions of arbitrary base forecasters by utilizing the implicit probability density learned by TSDiff as a prior.

**Problem Statement.** Let $\mathbf{y} \in \mathbb{R}^L$ be a time series of length $L$. Denote obs $\subset \{1, \ldots, L\}$ as the set of observed timesteps and ta as its complement set of target timesteps. Our goal is to recover the complete time series $\mathbf{y}$, given the observed subsequence $\mathbf{y}_{\text{obs}}$ which may or may not be contiguous. Formally, this involves modeling the conditional distribution $p_\theta(\mathbf{y}_{\text{ta}}|\mathbf{y}_{\text{obs}})$. This general setup subsumes forecasting tasks, with or without missing values, as special cases. We seek to train a single unconditional generative model, $p_\theta(\mathbf{y})$, and condition it during inference to draw samples from arbitrary distributions of interest, $p_\theta(\mathbf{y}_{\text{ta}}|\mathbf{y}_{\text{obs}})$.

**Generative Model Architecture.**
We begin with modeling the marginal probability, $p_\theta(\mathbf{y})$, via a diffusion model, referred to as TS-Diff, parameterized by $\theta$. The architecture of TSDiff is depicted in Fig. 2 and is based on SSSD [1] which is a modification of DiffWave [28] employing S4 layers [18]. TSDiff is designed to handle *univariate* sequences of length $L$. To incorporate historical information beyond $L$ timesteps without increasing $L$, we append lagged time series along the channel dimension. This results in a noisy input $\mathbf{x}^t \in \mathbb{R}^{L \times C}$ (see Eq. 2) to the diffusion model, where $C - 1$ is the number of lags. The S4 layers oper-

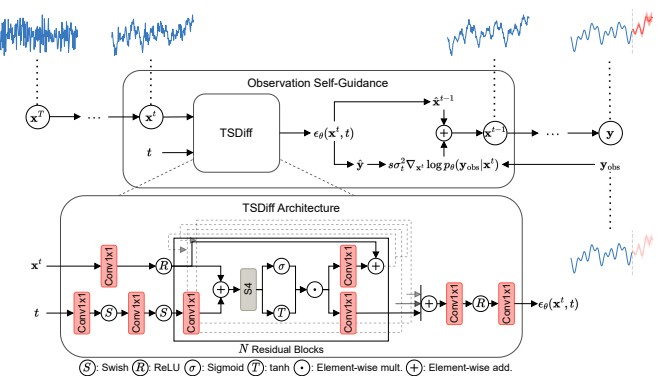

Figure 2: An overview of observation self-guidance. The predicted noise, $\boldsymbol{\epsilon}_\theta(\mathbf{x}^t, t)$, first denoises $\mathbf{x}^t$ unconditionally as $\hat{\mathbf{x}}^{t-1}$ and approximates $\mathbf{y}$ as $\hat{\mathbf{y}}$. The reverse diffusion step then guides $\hat{\mathbf{x}}^{t-1}$ via the log-likelihood of the observation $\mathbf{y}_{\text{obs}}$ under a distribution parameterized by $\hat{\mathbf{y}}$.

ate on the time dimension whereas the Conv1x1 layers operate on the channel dimension, facilitating information flow along both dimensions. As typical for unconditional diffusion models, the output dimensions of TSDiff match the input dimensions. Note that while we focus on univariate time series in this work, TSDiff can be modified to handle multivariate time series by incorporating additional layers, e.g., a transformer layer, operating across the feature dimensions after the S4 layer.

In the following, we discuss two approaches to condition the generative model, $p_\theta(\mathbf{y})$, during inference, enabling us to draw samples from $p_\theta(\mathbf{y}_{\text{ta}}|\mathbf{y}_{\text{obs}})$.

## 3.1 Observation Self-Guidance

Let $t \geq 0$ be an arbitrary diffusion step. Applying Bayes' rule, we have,

$$p_\theta(\mathbf{x}^t|\mathbf{y}_{\text{obs}}) \propto p_\theta(\mathbf{y}_{\text{obs}}|\mathbf{x}^t)p_\theta(\mathbf{x}^t), \tag{8}$$

which yields the following relation between the conditional and marginal score functions,

$$\nabla_{\mathbf{x}^t} \log p_\theta(\mathbf{x}^t|\mathbf{y}_{\text{obs}}) = \nabla_{\mathbf{x}^t} \log p_\theta(\mathbf{y}_{\text{obs}}|\mathbf{x}^t) + \nabla_{\mathbf{x}^t} \log p_\theta(\mathbf{x}^t). \tag{9}$$

Given access to the guidance distribution, $p_\theta(\mathbf{y}_{\text{obs}}|\mathbf{x}^t)$, we can draw samples from $p_\theta(\mathbf{y}_{\text{ta}}|\mathbf{y}_{\text{obs}})$ using guided reverse diffusion, akin to Eq. (7),

$$p_\theta(\mathbf{x}^{t-1}|\mathbf{x}^t, \mathbf{y}_{\text{obs}}) = \mathcal{N}(\mathbf{x}^{t-1}; \mu_\theta(\mathbf{x}^t, t) + s\sigma_t^2 \nabla_{\mathbf{x}^t} \log p_\theta(\mathbf{y}_{\text{obs}}|\mathbf{x}^t), \sigma_t^2 \mathbf{I}). \tag{10}$$

The scale parameter $s$ controls how strongly the observations, $\mathbf{y}_{\text{obs}}$, and the corresponding timesteps in the diffused time series, $\mathbf{y}$, align. Unlike Dhariwal and Nichol [10], we do not have access to auxiliary guidance networks. In the following, we propose two variants of a *self-guidance* mechanism that utilizes the same diffusion model to parameterize the guidance distribution. The main intuition behind self-guidance is that a model designed for complete sequences should reasonably approximate partial sequences. A pseudo-code of the observation self-guidance is given in App. A.2.

**Mean Square Self-Guidance.** We model $p_\theta(\mathbf{y}_{\text{obs}}|\mathbf{x}^t)$ as a multivariate Gaussian distribution,

$$p_\theta(\mathbf{y}_{\text{obs}}|\mathbf{x}^t) = \mathcal{N}(\mathbf{y}_{\text{obs}}|f_\theta(\mathbf{x}^t, t), \mathbf{I}), \tag{11}$$

where $f_\theta$ is a function approximating $\mathbf{y}$, given the noisy time series $\mathbf{x}^t$. We can reuse the denoising network $\boldsymbol{\epsilon}_\theta$ to estimate $\mathbf{y}$ as

$$\hat{\mathbf{y}} = f_\theta(\mathbf{x}^t, t) = \frac{\mathbf{x}^t - \sqrt{(1 - \bar{\alpha}_t)}\boldsymbol{\epsilon}_\theta(\mathbf{x}^t, t)}{\sqrt{\bar{\alpha}_t}}, \tag{12}$$

which follows by rearranging Eq. (2) with $\boldsymbol{\epsilon} = \boldsymbol{\epsilon}_\theta(\mathbf{x}^t, t)$, as shown by Song et al. [49]. This one-step denoising serves as a cost-effective approximation of the model for the observed time series and provides the requisite guidance term in the form of the score function, $\nabla_{\mathbf{x}^t} \log p_\theta(\mathbf{y}_{\mathrm{obs}}|\mathbf{x}^t)$, which can be computed by automatic differentiation. Consequently, our self-guidance approach requires no auxiliary networks or changes to the training procedure. Applying the logarithm to Eq. (11) and dropping constant terms yields the mean squared error (MSE) loss on the observed part of the time series, hence we named this technique mean square self-guidance.

**Quantile Self-Guidance.** Probabilistic forecasts are often evaluated using quantile-based metrics such as the continuous ranked probability score (CRPS) [15]. While the MSE only quantifies the average quadratic deviation from the mean, the CRPS takes all quantiles of the distribution into account by integrating the quantile loss (also known as the pinball loss) from 0 to 1. This motivated us to substitute the Gaussian distribution with the asymmetric Laplace distribution that has been studied in the context of Bayesian quantile regression [58]. The probability density function of the asymmetric Laplace distribution is given by,

$$p_\theta(\mathbf{y}_{\mathrm{obs}}|\mathbf{x}^t) = \frac{1}{Z} \cdot \exp\left(-\frac{1}{b}\max\left\{\kappa \cdot (\mathbf{y}_{\mathrm{obs}} - f_\theta(\mathbf{x}^t, t)), (\kappa - 1) \cdot (\mathbf{y}_{\mathrm{obs}} - f_\theta(\mathbf{x}^t, t))\right\}\right), \tag{13}$$

where $Z$ is a normalization constant, $b > 0$ a scale parameter, and $\kappa \in (0, 1)$ an asymmetry parameter. Setting $b = 1$, the log density yields the quantile loss with the score function,

$$\nabla_{\mathbf{x}^t} \log p_\theta(\mathbf{y}_{\mathrm{obs}}|\mathbf{x}^t) = \nabla_{\mathbf{x}^t} \max\{\kappa \cdot (\mathbf{y}_{\mathrm{obs}} - f_\theta(\mathbf{x}^t, t)), (\kappa - 1) \cdot (\mathbf{y}_{\mathrm{obs}} - f_\theta(\mathbf{x}^t, t))\}, \tag{14}$$

with $\kappa$ specifying the quantile level. By plugging Eq. (14) into Eq. (9), the reverse diffusion can be guided towards a specific quantile level $\kappa$. In practice, we use multiple evenly spaced quantile levels in $(0, 1)$, based on the number of samples in the forecast. Intuitively, we expect quantile self-guidance to generate more diverse predictions by better representing the cumulative distribution function.

## 3.2 Prediction Refinement

In the previous section, we discussed a technique enabling the unconditional model to generate predictions by employing diffusion guidance. In this section, we will discuss another approach repurposing the model to refine predictions of base forecasters. Our approach is completely agnostic to the type of base forecaster and only assumes access to forecasts generated by them. The initial forecasts are iteratively refined using the implicit density learned by the diffusion model which serves as a prior. Unlike reverse diffusion which requires sequential sampling of all latent variables, refinement is performed directly in the data space. This provides a trade-off between quality and computational overhead, making it an economical alternative when the number of refinement iterations is less than the number of diffusion steps. Furthermore, in certain industrial forecasting scenarios, one has access to a complex production forecasting system of black-box nature. In these cases, refinement presents a cost-effective solution that enhances forecast accuracy post hoc, without modifying the core forecasting process — a change that could potentially be a lengthy procedure.

In the following, we present two interpretations of refinement as (a) sampling from an energy function, and (b) maximizing the likelihood to find the most likely sequence.

**Energy-Based Sampling.** Recall that our goal is to draw samples from the distribution $p(\mathbf{y}_{\mathrm{ta}}|\mathbf{y}_{\mathrm{obs}})$. Let $g$ be an arbitrary base forecaster and $g(\mathbf{y}_{\mathrm{obs}})$ be a sample forecast from $g$ which serves as an initial guess of a sample from $p(\mathbf{y}_{\mathrm{ta}}|\mathbf{y}_{\mathrm{obs}})$. To improve this initial guess, we formulate refinement as the problem of sampling from the regularized energy-based model (EBM),

$$E_\theta(\mathbf{y}; \tilde{\mathbf{y}}) = -\log p_\theta(\mathbf{y}) + \lambda \mathcal{R}(\mathbf{y}, \tilde{\mathbf{y}}), \tag{15}$$

where $\tilde{\mathbf{y}}$ is the time series obtained upon combining $\mathbf{y}_{\mathrm{obs}}$ and $g(\mathbf{y}_{\mathrm{obs}})$, and $\mathcal{R}$ is a regularizer such as the MSE loss or the quantile loss. We designed the energy function such that low energy is assigned to samples that are likely under the diffusion model, $p_\theta(\mathbf{y})$, and also close to $\tilde{\mathbf{y}}$, ensured by the diffusion log-likelihood and the regularizer in the energy function, respectively. The Lagrange

multiplier, $\lambda$, may be tuned to control the strength of regularization; however, we set it to $1$ in our experiments for simplicity.

We use *overdamped* Langevin Monte Carlo (LMC) [50] to sample from this EBM. $\mathbf{y}_{(0)}$ is initialized to $\tilde{\mathbf{y}}$ and iteratively refined as,

$$\mathbf{y}_{(i+1)} = \mathbf{y}_{(i)} - \eta \nabla_{\mathbf{y}_{(i)}} E_\theta(\mathbf{y}_{(i)}; \tilde{\mathbf{y}}) + \sqrt{2\eta\gamma}\xi_i \quad \text{and} \quad \xi_i \sim \mathcal{N}(\mathbf{0}, \mathbf{I}), \tag{16}$$

where $\eta$ and $\gamma$ are the step size and noise scale, respectively.

Note that in contrast to observation self-guidance, we directly refine the time series in the data space and require an initial forecast from a base forecaster. However, similar to observation self-guidance, this approach does not require any modifications to the training procedure and can be applied to any trained diffusion model. A pseudo-code of the energy-based refinement is provided in App. A.2.

**Maximizing the Likelihood.** The decomposition in Eq. (15) can also be interpreted as a regularized optimization problem with the goal of finding the most likely time series that satisfies certain constraints on the observed timesteps. Concretely, it translates into,

$$\arg\min_{\mathbf{y}} \left[ -\log p_\theta(\mathbf{y}) + \lambda \mathcal{R}(\mathbf{y}, \tilde{\mathbf{y}}) \right], \tag{17}$$

which can be optimized using gradient descent and is a special case of Eq. (16) with $\gamma = 0$. Given the non-convex nature of this objective, convergence to the global optimum is not guaranteed. Therefore, we expect the initial value, $\mathbf{y}_{(0)}$, to influence the resulting time series, which can also be observed in the experiment results (see Table 3).

**Approximation of** $\log p_\theta(\mathbf{y})$**.** To approximate the log-likelihood $\log p_\theta(\mathbf{y})$ we can utilize the objective used to train diffusion models,

$$\log p_\theta(\mathbf{y}) \approx -\mathbb{E}_{\boldsymbol{\epsilon}, t} \left[ \|\boldsymbol{\epsilon}_\theta(\mathbf{x}^t, t) - \boldsymbol{\epsilon}\|^2 \right], \tag{18}$$

which is a simplification of the ELBO [20]. However, a good approximation requires sampling several diffusion steps ($t$) incurring computational overhead and slowing down inference. To speed up inference, we propose to approximate Eq. (18) using only a single diffusion step. Instead of randomly sampling $t$, we use the *representative step*, $\tau$, which improved refinement stability in our experiments. The representative step corresponds to the diffusion step that best approximates Eq. (18), i.e.,

$$\tau = \arg\min_{\tilde{t}} \left( \mathbb{E}_{\boldsymbol{\epsilon}, t, \mathbf{y}} \left[ \|\boldsymbol{\epsilon}_\theta(\mathbf{x}^t, t) - \boldsymbol{\epsilon}\|^2 \right] - \mathbb{E}_{\boldsymbol{\epsilon}, \mathbf{y}} \left[ \|\boldsymbol{\epsilon}_\theta(\mathbf{x}^{\tilde{t}}, \tilde{t}) - \boldsymbol{\epsilon}\|^2 \right] \right)^2. \tag{19}$$

The representative step is computed only once per dataset. It can be efficiently computed post training by computing the loss at every diffusion step on a randomly-sampled batch of training datapoints and then finding the diffusion step closest to the average loss. Alternatively, we can keep a running average loss for each $t$ and compute the representative step using these running averages. This can be done efficiently because the loss used to obtain $\tau$ is the same as the training loss for the diffusion model.

## 4 Experiments

In this section, we present empirical results on several real-world datasets. Our goal is to investigate whether unconditional time series diffusion models can be employed for downstream tasks that typically require conditional models. Concretely, we tested if (a) the self-guidance mechanism in TSDiff can generate probabilistic forecasts (also in the presence of missing values), (b) the implicit probability density learned by TSDiff can be leveraged to refine the predictions of base forecasters, and (c) the synthetic samples generated by TSDiff are adequate for training downstream forecasters.[4]

**Datasets and Evaluation.** We conducted experiments on eight *univariate* time series datasets from different domains, available in GluonTS [2] — Solar [29], Electricity [11], Traffic [11], Exchange [29], M4-Hourly [37], UberTLC-Hourly [13], KDDCup [16], and Wikipedia [14]. We evaluated the quality of probabilistic forecasts using the *continuous ranked probability score* (CRPS) [15]. We approximated the CRPS by the normalized average quantile loss using 100 sample paths, and report means and standard deviations over three independent runs (see App. B for details on the datasets and evaluation metric).

---

[4]Our code is available at: `github.com/amazon-science/unconditional-time-series-diffusion`

Table 1: Forecasting results on eight benchmark datasets. The best and second best models have been shown as **bold** and underlined, respectively.

| Method | Solar | Electricity | Traffic | Exchange | M4-Hourly | UberTLC-Hourly | KDDCup | Wikipedia |
|---|---|---|---|---|---|---|---|---|
| Seasonal Naive | 0.512±0.000 | 0.069±0.000 | 0.221±0.000 | 0.011±0.000 | 0.048±0.000 | 0.299±0.000 | 0.561±0.000 | 0.410±0.000 |
| ARIMA | 0.545±0.006 | - | - | 0.008±0.000 | 0.044±0.001 | 0.284±0.001 | 0.547±0.003 | - |
| ETS | 0.611±0.040 | 0.072±0.004 | 0.433±0.050 | 0.008±0.000 | 0.042±0.001 | 0.422±0.001 | 0.753±0.008 | 0.715±0.002 |
| Linear | 0.569±0.021 | 0.088±0.008 | 0.179±0.003 | 0.011±0.001 | 0.039±0.001 | 0.360±0.023 | 0.513±0.011 | 1.624±1.114 |
| DeepAR | 0.389±0.001 | 0.054±0.000 | 0.099±0.001 | 0.011±0.003 | 0.052±0.006 | **0.161±0.002** | 0.414±0.027 | 0.231±0.008 |
| MQ-CNN | 0.790±0.063 | 0.067±0.001 | - | 0.019±0.006 | 0.046±0.003 | 0.436±0.020 | 0.516±0.012 | 0.220±0.001 |
| DeepState | 0.379±0.002 | 0.075±0.004 | 0.146±0.018 | 0.011±0.001 | 0.041±0.002 | 0.288±0.087 | - | 0.318±0.019 |
| Transformer | 0.419±0.008 | 0.076±0.018 | 0.102±0.002 | 0.010±0.000 | 0.040±0.014 | 0.192±0.004 | 0.411±0.021 | **0.214±0.001** |
| TFT | 0.417±0.023 | 0.086±0.008 | 0.134±0.007 | **0.007±0.000** | 0.039±0.001 | 0.193±0.006 | 0.581±0.053 | 0.229±0.006 |
| CSDI | 0.352±0.005 | 0.054±0.000 | 0.159±0.002 | 0.033±0.014 | 0.040±0.003 | 0.206±0.002 | 0.318±0.002 | 0.289±0.017 |
| TSDiff-Cond | **0.338±0.014** | 0.050±0.002 | **0.094±0.003** | 0.013±0.002 | 0.039±0.006 | 0.172±0.008 | 0.754±0.007 | 0.218±0.010 |
| TSDiff-MS | 0.391±0.003 | 0.062±0.001 | 0.116±0.001 | 0.018±0.003 | 0.045±0.000 | 0.183±0.007 | 0.325±0.028 | 0.257±0.001 |
| TSDiff-Q | 0.358±0.020 | **0.049±0.000** | 0.098±0.002 | 0.011±0.001 | **0.036±0.001** | 0.172±0.005 | **0.311±0.026** | 0.221±0.001 |

## 4.1 Forecasting using Observation Self-Guidance

We tested the forecasting performance of the two proposed variants of observation self-guidance, mean square guidance (TSDiff-MS) and quantile guidance (TSDiff-Q), against several forecasting baselines. We included Seasonal Naive, ARIMA, ETS, and a Linear (ridge) regression model from the statistical literature [21]. Additionally, we compared against deep learning models that represent various architectural paradigms such as the RNN-based DeepAR [46], the CNN-based MQ-CNN [54], the state space model-based DeepState [42], and the self-attention-based Transformer [53] and TFT [32] models. We also compared against two conditional diffusion models, CSDI [52] and TSDiff-Cond, a conditional version of TSDiff closely related to SSSD [1] (see App. B.4 for a more in-depth discussion on the baselines). Note that we did not seek to obtain state-of-the-art forecasting results on the datasets studied but to demonstrate the efficacy of unconditional diffusion models against task-specific conditional models.

Table 1 shows that TSDiff-Q is competitive with state-of-the-art conditional models, but does not require task-specific training, achieving the lowest or second-lowest CRPS on 5/8 datasets. We also observe that the choice of guidance distribution is critical. While Gaussian (TSDiff-MS) yields reasonable results, using the asymmetric Laplace distribution (TSDiff-Q) lowers the CRPS further. We hypothesize that taking the different quantile levels into account during guidance improves the results on the quantile-based evaluation metric (CRPS). Fig. 3 shows example forecasts from TSDiff-Q. Note that as self-guidance only imposes a soft-constraint on the observed timesteps, the resultant diffused values are not guaranteed to match the observations for these timesteps. The alignment between predictions and observations can be controlled via the scale parameter, $s$ (see Eq. 9). In practice we observed that the diffused values were close to the observations for our selected values of $s$, as shown in Fig. 5 (Appendix).

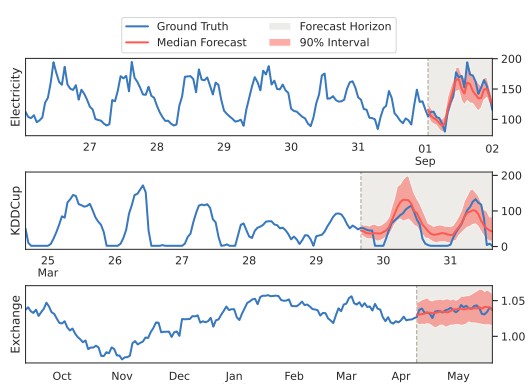

Figure 3: Example forecasts generated by TSDiff-Q for time series in Electricity, KDDCup, and Exchange — three datasets with different frequencies and/or prediction lengths.

**Forecasting with Missing Values.** We evaluated the forecasting performance with missing values in the historical context during inference to demonstrate the flexibility of self-guidance. Specifically, we tested three scenarios: (a) random missing (RM), (b) blackout missing (i.e., a window with consecutive missing values) at the beginning of the context window (BM-B), and (c) blackout missing at the end of the context window (BM-E). We removed a segment equal to the context window from the end of the training time series to ensure that the model is not trained on this section. During inference, we masked 50% of the timesteps from this held-out context window for each scenario

while the forecast horizon remained unchanged. We trained the unconditional model on complete sequences only once per dataset, without making any modifications to the objective function. As a result, the three scenarios described above are *inference only* for TSDiff. For comparison, we trained conditional models (TSDiff-Cond) specifically on the target missingness scenarios. These models were trained to predict the missing values in the input time series, where some timesteps were masked according to the target missingness scenario (see App. B.3 for a detailed discussion on the missing values experiment setup).

Table 2: Forecasting with missing values results on six benchmark datasets.

|  | Method | Solar | Electricity | Traffic | Exchange | UberTLC-Hourly | KDDCup |
|---|---|---|---|---|---|---|---|
| RM | TSDiff-Cond | 0.357±0.023 | 0.052±0.001 | 0.097±0.003 | 0.012±0.004 | 0.180±0.015 | 0.757±0.024 |
| RM | TSDiff-Q | 0.387±0.015 | 0.052±0.001 | 0.110±0.004 | 0.013±0.000 | 0.183±0.002 | 0.397±0.042 |
| BM-B | TSDiff-Cond | 0.377±0.017 | 0.049±0.001 | 0.094±0.005 | 0.009±0.000 | 0.181±0.009 | 0.699±0.009 |
| BM-B | TSDiff-Q | 0.387±0.019 | 0.051±0.000 | 0.110±0.006 | 0.011±0.001 | 0.182±0.004 | 0.441±0.096 |
| BM-E | TSDiff-Cond | 0.376±0.036 | 0.065±0.003 | 0.123±0.023 | 0.035±0.021 | 0.179±0.013 | 0.819±0.033 |
| BM-E | TSDiff-Q | 0.435±0.113 | 0.068±0.009 | 0.139±0.013 | 0.020±0.001 | 0.183±0.005 | 0.344±0.012 |

The results in Table 2 show that TSDiff-Q performs competitively against task-specific conditional models, demonstrating its robustness with respect to missing values during inference. This makes TSDiff-Q particularly useful in real-world applications where missing values are common but the missingness scenario is not known in advance. It is noteworthy that we did not tune the guidance scale for each missingness scenario but used the same value as in the previous standard forecasting setup. We expect the results to further improve if the scale is carefully updated based on the missingness ratio and scenario.

## 4.2 Refining Predictions of Base Forecasters

We evaluated the quality of the implicit probability density learned by TSDiff by refining predictions from base forecasters, as described in Sec. 3.2. We considered two point forecasters (Seasonal Naive and Linear) and two probabilistic forecasters (DeepAR and Transformer) as base models. We refine the forecasts generated from these base models for 20 steps, as described in Algorithm 2 (see App. B.3 for a discussion on the number of refinement steps). Similar to self-guidance, we experimented with the Gaussian and asymmetric Laplace negative log-likelihoods as regularizers, $\mathcal{R}$, for both energy (denoted by LMC) and maximum likelihood (denoted by ML)-based refinement.

The CRPS scores before (denoted as "Base") and after refinement are reported in Table 3. At least one refinement method reduces the CRPS of the base forecast on every dataset. Energy-based refinement generally performs better than ML-based refinement for point forecasters. We hypothesize that this is

Table 3: Refinement results on eight benchmark datasets. The best and second best settings have been shown as **bold** and underlined, respectively.

|  | Setting | Solar | Electricity | Traffic | Exchange | M4-Hourly | UberTLC-Hourly | KDDCup | Wikipedia |
|---|---|---|---|---|---|---|---|---|---|
| Seasonal Naive | Base | 0.512±0.000 | 0.069±0.000 | 0.221±0.000 | 0.011±0.000 | 0.048±0.000 | 0.299±0.000 | 0.561±0.000 | 0.410±0.000 |
| Seasonal Naive | LMC-MS | **0.480±0.009** | 0.059±0.004 | **0.126±0.001** | 0.013±0.001 | 0.040±0.002 | **0.186±0.005** | 0.505±0.027 | **0.339±0.001** |
| Seasonal Naive | LMC-Q | **0.480±0.007** | 0.051±0.001 | 0.134±0.004 | **0.009±0.000** | 0.036±0.001 | 0.204±0.007 | **0.399±0.003** | 0.357±0.001 |
| Seasonal Naive | ML-MS | 0.489±0.007 | 0.064±0.006 | 0.130±0.002 | 0.015±0.002 | 0.046±0.003 | 0.202±0.004 | 0.519±0.028 | 0.349±0.001 |
| Seasonal Naive | ML-Q | **0.480±0.007** | 0.050±0.001 | 0.135±0.004 | 0.009±0.000 | 0.036±0.001 | 0.215±0.008 | 0.403±0.003 | 0.365±0.001 |
| Linear | Base | 0.569±0.021 | 0.088±0.008 | 0.179±0.003 | 0.011±0.001 | 0.039±0.001 | 0.360±0.023 | 0.513±0.011 | 1.624±1.114 |
| Linear | LMC-MS | **0.494±0.019** | 0.059±0.004 | **0.113±0.001** | 0.013±0.001 | 0.040±0.002 | **0.187±0.007** | 0.458±0.015 | **1.315±0.992** |
| Linear | LMC-Q | 0.516±0.020 | **0.055±0.003** | 0.119±0.002 | **0.009±0.000** | 0.034±0.001 | 0.228±0.010 | **0.346±0.010** | 1.329±1.002 |
| Linear | ML-MS | 0.503±0.016 | 0.063±0.005 | 0.117±0.002 | 0.015±0.002 | 0.045±0.003 | 0.203±0.007 | 0.472±0.015 | 1.327±0.993 |
| Linear | ML-Q | 0.523±0.021 | 0.056±0.003 | 0.121±0.003 | 0.010±0.001 | **0.032±0.001** | 0.240±0.010 | 0.350±0.011 | 1.335±1.002 |
| DeepAR | Base | 0.389±0.001 | 0.054±0.000 | **0.099±0.001** | 0.011±0.003 | 0.052±0.006 | 0.161±0.002 | 0.414±0.027 | 0.231±0.008 |
| DeepAR | LMC-MS | 0.398±0.004 | 0.059±0.004 | 0.111±0.001 | 0.012±0.001 | 0.040±0.002 | 0.184±0.005 | 0.469±0.034 | 0.227±0.002 |
| DeepAR | LMC-Q | 0.388±0.002 | 0.053±0.001 | 0.101±0.001 | **0.010±0.001** | 0.035±0.001 | 0.161±0.002 | **0.401±0.021** | **0.220±0.005** |
| DeepAR | ML-MS | 0.402±0.009 | 0.064±0.006 | 0.115±0.002 | 0.014±0.001 | 0.046±0.003 | 0.198±0.005 | 0.477±0.034 | 0.235±0.002 |
| DeepAR | ML-Q | **0.386±0.002** | 0.052±0.001 | **0.099±0.001** | 0.010±0.002 | 0.035±0.001 | **0.160±0.002** | 0.401±0.021 | 0.221±0.006 |
| Transformer | Base | 0.419±0.008 | 0.076±0.018 | 0.102±0.002 | **0.010±0.000** | 0.040±0.014 | 0.192±0.004 | 0.411±0.021 | 0.214±0.001 |
| Transformer | LMC-MS | 0.415±0.009 | 0.059±0.004 | 0.111±0.001 | 0.013±0.001 | 0.040±0.002 | 0.185±0.005 | 0.462±0.014 | 0.229±0.003 |
| Transformer | LMC-Q | 0.415±0.008 | **0.058±0.003** | 0.101±0.001 | **0.010±0.000** | 0.038±0.006 | **0.177±0.005** | **0.384±0.005** | 0.211±0.002 |
| Transformer | ML-MS | 0.418±0.010 | 0.063±0.005 | 0.115±0.002 | 0.014±0.002 | 0.046±0.003 | 0.198±0.005 | 0.470±0.014 | 0.238±0.003 |
| Transformer | ML-Q | **0.413±0.008** | 0.059±0.005 | **0.099±0.001** | **0.010±0.000** | 0.037±0.006 | **0.177±0.005** | 0.384±0.006 | **0.210±0.002** |

Table 4: Results of forecasters trained on synthetic samples from different generative models on eight benchmark datasets. Best scores are shown in **bold**.

| | Generator | Solar | Electricity | Traffic | Exchange | M4-Hourly | UberTLC-Hourly | KDDCup | Wikipedia |
|---|---|---|---|---|---|---|---|---|---|
| **Linear (LPS)** | Real | 0.569±0.021 | 0.088±0.008 | 0.179±0.003 | 0.011±0.001 | 0.039±0.001 | 0.360±0.023 | 0.513±0.011 | 1.624±1.114 |
| | TimeVAE | 0.933±0.147 | 0.128±0.005 | 0.236±0.010 | 0.024±0.004 | 0.074±0.003 | 0.354±0.020 | 1.020±0.179 | 0.643±0.068 |
| | TimeGAN | 1.140±0.583 | 0.234±0.064 | 0.398±0.092 | **0.011±0.000** | 0.140±0.053 | 0.665±0.104 | 0.713±0.009 | 0.421±0.023 |
| | TSDiff | **0.581±0.032** | **0.065±0.002** | **0.164±0.002** | 0.012±0.001 | **0.045±0.007** | **0.291±0.084** | **0.481±0.013** | **0.392±0.013** |
| **DeepAR** | Real | 0.389±0.001 | 0.054±0.000 | 0.099±0.001 | 0.011±0.003 | 0.052±0.006 | 0.161±0.002 | 0.414±0.027 | 0.231±0.008 |
| | TimeVAE | 0.493±0.012 | 0.060±0.001 | 0.155±0.006 | 0.009±0.000 | **0.039±0.010** | 0.278±0.009 | 0.621±0.003 | 0.440±0.012 |
| | TimeGAN | 0.976±0.739 | 0.183±0.036 | 0.419±0.122 | **0.008±0.001** | 0.121±0.035 | 0.594±0.125 | 0.690±0.091 | 0.322±0.048 |
| | TSDiff | **0.478 ±0.007** | **0.058±0.001** | **0.129±0.003** | 0.017±0.009 | 0.042±0.024 | **0.191±0.018** | **0.378±0.012** | **0.222±0.005** |
| **Transf.** | Real | 0.419±0.008 | 0.076±0.018 | 0.102±0.002 | 0.010±0.000 | 0.040±0.014 | 0.192±0.004 | 0.411±0.021 | 0.214±0.001 |
| | TimeVAE | 0.520±0.030 | 0.071±0.009 | 0.163±0.018 | 0.011±0.001 | 0.035±0.011 | 0.291±0.008 | 0.717±0.181 | 0.451±0.017 |
| | TimeGAN | 0.972±0.687 | 0.182±0.008 | 0.413±0.204 | **0.009±0.001** | 0.114±0.052 | 0.685±0.448 | 0.632±0.016 | 0.314±0.045 |
| | TSDiff | **0.457±0.008** | **0.056±0.001** | **0.143±0.020** | 0.030±0.021 | **0.030±0.008** | **0.225±0.055** | **0.356±0.030** | **0.239±0.010** |

because the additive noise in LMC incentivizes exploration of the energy landscape which improves the "spread" of the initial point forecast. In contrast, ML-Q refinement outperforms energy-based refinement on probabilistic base models suggesting that sampling might not be necessary for these models due to the probabilistic nature of the initial forecasts.

Refinement provides a trade-off; while it has a lower computational overhead than self-guidance, its performance strongly depends on the chosen base forecaster. It substantially improves the performance of simple point forecasters (e.g., Seasonal Naive) and yields improvements on stronger probabilistic forecasters (e.g., Transformer) as well.

## 4.3 Training Downstream Models using Synthetic Data

Finally, we evaluated the quality of generated samples through the forecasting performance of downstream models trained on these synthetic samples. Several metrics have previously been proposed to evaluate the quality of synthetic samples [12, 57, 26, 23]. Of these, we primarily focused on *predictive* metrics that involve training a downstream model on synthetic samples and evaluating its performance on real data (i.e., the *train on synthetic-test on real* setup). In the absence of a standard downstream model (such as the Inception network [51] in the case of images), prior works have proposed different network architectures for the downstream model. This makes the results sensitive to architecture choice, random initialization, and even the choice of deep learning library. Furthermore, training a downstream model introduces additional overhead per metric computation. We propose the Linear Predictive Score (LPS) to address these issues. We define the LPS as the test CRPS of a linear (ridge) regression model trained on the synthetic samples. The ridge regression model is a simple, standard model available in standard machine learning libraries (e.g., `scikit-learn`) that can effectively gauge the predictive quality of synthetic samples. Moreover, a ridge regression model is cheap to fit — it can be solved in closed-form, eliminating variance introduced by initialization and training.

We compared samples generated by TSDiff against those from TimeGAN [57] and TimeVAE [9], two time series generative models from alternative frameworks. In addition to the linear model, we trained two strong downstream forecasters (DeepAR and Transformer) on synthetic samples from each generative model. The test CRPS of these forecasters is presented in Table 4. TSDiff's samples significantly outperform TimeVAE and TimeGAN in terms of the LPS showcasing their excellent predictive quality with respect to a simple (linear) forecaster. On the stronger DeepAR and Transformer forecasters, TSDiff outperforms the baselines on most of the datasets. Moreover, the scores obtained by TSDiff are reasonable when compared to those attained by downstream forecasters trained on real samples. Notably, these forecasters trained on real data had the advantage of accessing time features and lags that extend far beyond the sequence length modeled by TSDiff. These results serve as evidence that TSDiff effectively captures crucial patterns within time series datasets and generates realistic samples (see App. C for a qualitative comparison between real time series and those generated by TSDiff).

# 5    Related Work

**Diffusion Models.**    Diffusion models were originally proposed for image synthesis [48, 20, 10], but have since been applied to other domains such as audio synthesis [28], protein modeling [3], and graph modeling [24]. Prior works have also studied image inpainting using diffusion models [35, 45, 25] which is a problem related to time series imputation. Similar to the maximum likelihood variant of our refinement approach, Graikos et al. [17] showed the utility of diffusion models as plug-and-play priors. The architecture of our proposed model, TSDiff, is based on a modification [1] of DiffWave [28] which was originally developed for audio synthesis.

**Diffusion Models for Time Series.**    Conditional diffusion models have been applied to time series tasks such as imputation and forecasting. The first work is due to Rasul et al. [44], who proposed TimeGrad for multivariate time series forecasting featuring a conditional diffusion head. Biloš et al. [6] extended TimeGrad to continuous functions and made modifications to the architecture enabling simultaneous multi-horizon forecasts. CDSI [52] is a conditional diffusion model for imputation and forecasting which is trained by masking the observed time series with different strategies. SSSD [1] is a modification of CDSI that uses S4 layers [18] as the fundamental building block instead of transformers. The models discussed above [44, 52, 1] are all conditional models, i.e., they are trained on specific imputation or forecasting tasks. In contrast, TSDiff is trained unconditionally and conditioned during inference using diffusion guidance, making it applicable to various downstream tasks.

**Diffusion Guidance.**    Dhariwal and Nichol [10] proposed classifier guidance to repurpose pretrained unconditional diffusion models for conditional image generation using auxiliary image classifiers. Ho and Salimans [19] eliminated the need for an additional classifier by jointly training a conditional and an unconditional diffusion model and combining their score functions for conditional generation. Diffusion guidance has also been employed for text-guided image generation and editing [39, 4] using CLIP embeddings [41]. In contrast to prior work, our proposed observation self-guidance does not require auxiliary networks or modifications to the training procedure. Furthermore, we apply diffusion guidance to the time series domain while previous works primarily focus on images.

# 6    Conclusion

In this work, we proposed TSDiff, an unconditional diffusion model for time series, and a self-guidance mechanism that enables conditioning TSDiff for probabilistic forecasting tasks during inference, without requiring auxiliary guidance networks or modifications to the training procedure. We demonstrated that our task-agnostic TSDiff, in combination with self-guidance, is competitive with strong task-specific baselines on multiple forecasting tasks. Additionally, we presented a refinement scheme to improve predictions of base forecasters by leveraging the implicit probability density learned by TSDiff. Finally, we validated its generative modeling capabilities by training multiple downstream forecasters on synthetic samples generated by TSDiff. Samples from TSDiff outperform alternative generative models in terms of their predictive quality. Our results indicate that TSDiff learns important characteristics of time series datasets, enabling conditioning during inference and high-quality sample generation, offering an attractive alternative to task-specific conditional models.

**Limitations and Future Work.**    While observation self-guidance provides an alternative approach for state-of-the-art probabilistic time series forecasting, its key limitation is the high computational cost of the iterative denoising process. Faster diffusion solvers [49, 33] would provide a trade-off between predictive performance and computational overhead without altering the training procedure. Furthermore, solvers developed specifically for accelerated guided diffusion [34, 55] could be deployed to speed up sampling without sacrificing predictive performance. Forecast refinement could be further improved by better approximations of the probability density and the utilization of momentum-based MCMC techniques such as Hamiltonian Monte Carlo [38] and underdamped Langevin Monte Carlo [50]. In this work, we evaluated TSDiff in the context of probabilistic forecasting, however, it is neither limited nor tailored to it. It can be easily applied to any imputation task similar to our forecasting with missing values experiment where we only evaluated the forecasting performance. Moreover, the core idea behind observation self-guidance is not limited to time series and may be applicable to other domains. Tailoring the guidance distribution appropriately, enables the use of self-guidance to a variety of inverse problems.

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

# A Technical Details

## A.1 Asymmetric Laplace Distribution

The asymmetric Laplace distribution is a generalization of the (symmetric) Laplace distribution and can be viewed as two exponential distributions with unequal rates glued about a location parameter. An alternative parameterization of this distribution has been employed for Bayesian quantile regression [58],

$$
p_\theta(z) \propto \exp\left(-\frac{1}{b}\max\{\kappa \cdot (z - m), (\kappa - 1) \cdot (z - m)\}\right),
$$

where $\kappa$ is the quantile level, $m$ is the location parameter, and $b$ is the scale parameter. Setting $\kappa = 0.5$ yields the symmetric Laplace distribution. The negative log-likelihood of the asymmetric Laplace distribution corresponds to the pinball loss. An illustration of the probability density function (PDF) and the unnormalized negative log-likelihood (NLL) for different quantile levels is shown in Fig. 4.

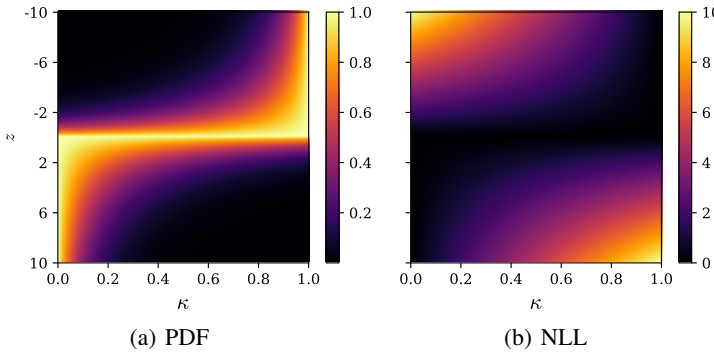

(a) PDF  (b) NLL

Figure 4: Probability density function (PDF) and negative log-likelihood (NLL) of the unnormalized Asymmetric Laplace Distribution for different quantile levels ($\kappa$) with $m = 0$ and $b = 1$.

## A.2 Algorithms

**Observation Self-Guidance.** The observation self-guidance can be easily implemented using guided reverse diffusion [10]. The pseudo-code of observation self-guidance is provided in Alg. 1. Given an observation, $\mathbf{y}_{\mathrm{obs}}$, and guidance scale, $s$, the algorithm starts from Gaussian noise, $\mathbf{x}^T$, which is iteratively denoised while being guided towards the observation.

---

**Algorithm 1** Observation Self-Guidance

**Input:** observation $\mathbf{y}_{\mathrm{obs}}$, scale $s$
$\mathbf{x}^T \sim \mathcal{N}(\mathbf{0}, \mathbf{I})$
**for** $t = T$ **to** $1$ **do**
    $\mathbf{x}^{t-1} \sim \mathcal{N}(\mu_\theta(\mathbf{x}^t, t), \mathbf{I}) + s\sigma_t^2 \nabla_{\mathbf{x}^t} \log p(\mathbf{y}_{\mathrm{obs}}|\mathbf{x}^t)$
**end for**

---

**Refinement.** The energy-based refinement scheme requires an observation, $\mathbf{y}_{\mathrm{obs}}$, a base forecaster, $g$, a step size, $\eta$, and a noise factor, $\gamma$. The pseudo-code of the energy-based refinement scheme is shown in Alg. 2. First, $\mathbf{y}_{\mathrm{obs}}$ and the prediction of the base forecaster, $g(\mathbf{y}_{\mathrm{obs}})$, are combined to $\tilde{\mathbf{y}}$ and used as initial guess, $\mathbf{y}_{(0)}$. This initial guess is then iteratively refined using the score function of the conditional distribution, $p_\theta(\mathbf{y}_{(i)}|\tilde{\mathbf{y}})$. Setting $\gamma = 0$ in Alg. 2 yields the maximum likelihood refinement scheme.

**Algorithm 2** Energy-Based Refinement

---

**Input:** observation $\mathbf{y}_{\mathrm{obs}}$, base forecaster $g$, step size $\eta$, noise factor $\gamma$, number of iterations $N$
$\tilde{\mathbf{y}} \leftarrow \mathrm{Combine}(\mathbf{y}_{\mathrm{obs}}, g(\mathbf{y}_{\mathrm{obs}}))$
$\mathbf{y}_{(0)} \leftarrow \tilde{\mathbf{y}}$
**for** $i = 0$ **to** $N - 1$ **do**
$\quad \xi_i \sim \mathcal{N}(0, \mathbf{I})$
$\quad \mathbf{y}_{(i+1)} \leftarrow \mathbf{y}_{(i)} - \eta \nabla_{\mathbf{y}_{(i)}} \log p_\theta(\mathbf{y}_{(i)} | \tilde{\mathbf{y}}) + \sqrt{2\eta\gamma} \xi_i$
**end for**

---

# B    Experiment Details

## B.1    Datasets

We used eight popular univariate datasets from different domains in our experiments. Preprocessed versions of these datasets are available in GluonTS [2] with associated frequencies (daily or hourly) and prediction lengths. Table 5 shows an overview of the datasets. We used sequence lengths ($L$, Context Length + Prediction Length) of 360 (i.e., 15 days) for the hourly datasets and 390 (i.e., approximately 13 months) for the daily datasets. These sequences were generated by slicing the original time series at random timesteps. In the following, we briefly describe the individual datasets.

- **Solar** [29] is a dataset of the photo-voltaic (i.e., solar) power production in the year 2006 from 137 solar power plants in the state of Alabama.[5]
- **Electricity** [29, 11] is a dataset of the electricity consumption of 370 customers.[6]
- **Traffic** [29, 11] is a dataset of the hourly occupancy rates on the San Francisco Bay area freeways from 2015 to 2016.[7]
- **Exchange** [29] consists of daily exchange rates of eight countries including Australia, British, Canada, Switzerland, China, Japan, New Zealand, and Singapore from 1990 to 2016.[8]
- **M4-Hourly** [37] is the hourly subset from the M4 competition.[9]
- **KDDCup** [16] is a dataset of the air quality indices (AQIs) of Beijing and London used in the KDD Cup 2018.[10]
- **UberTLC-Hourly** [13] is a dataset of Uber pickups from Jan to Jun 2015 obtained from the NYC Taxi & Limousine Commission (TLC).[11]
- **Wikipedia** [14] is a dataset of daily count of the number of hits on 2000 Wikipedia pages.[12]

Table 5: Overview of the benchmark datasets used in our experiments.

| Dataset | GluonTS Name | Train Size | Test Size | Domain | Freq. | Median Seq. Length | Context Length | Prediction Length |
|---|---|---|---|---|---|---|---|---|
| Solar | solar_nips | 137 | 959 | $\mathbb{R}^+$ | H | 7009 | 336 | 24 |
| Electricity | electricity_nips | 370 | 2590 | $\mathbb{R}^+$ | H | 5833 | 336 | 24 |
| Traffic | traffic_nips | 963 | 6741 | $(0, 1)$ | H | 4001 | 336 | 24 |
| Exchange | exchange_rate_nips | 8 | 40 | $\mathbb{R}^+$ | D | 6071 | 360 | 30 |
| M4-Hourly | m4_hourly | 414 | 414 | $\mathbb{N}$ | H | 960 | 312 | 48 |
| KDDCup | kdd_cup_2018_without_missing | 270 | 270 | $\mathbb{N}$ | H | 10850 | 312 | 48 |
| UberTLC-Hourly | uber_tlc_hourly | 262 | 262 | $\mathbb{N}$ | H | 4320 | 336 | 24 |
| Wikipedia | wiki2000_nips | 2000 | 10000 | $\mathbb{N}$ | D | 792 | 360 | 30 |

---

[5]Solar: https://www.nrel.gov/grid/solar-power-data.html

[6]Electricity: https://archive.ics.uci.edu/ml/datasets/ElectricityLoadDiagrams20112014

[7]Traffic: https://zenodo.org/record/4656132

[8]Exchange: https://github.com/laiguokun/multivariate-time-series-data

[9]M4-Hourly: https://github.com/Mcompetitions/M4-methods/tree/master/Dataset

[10]KDDCup: https://zenodo.org/record/4656756

[11]UberTLC-Hourly: https://github.com/fivethirtyeight/uber-tlc-foil-response

[12]Wikipedia: https://github.com/mbohlkeschneider/gluon-ts/tree/mv_release/datasets

## B.2 Metrics

**Continuous Ranked Probability Score (CRPS).** The CRPS is a popular metric used to evaluate the quality of probabilistic forecasts. It is a proper scoring rule [15] and is defined as the integral of the pinball loss from 0 to 1, i.e.,

$$\text{CRPS}(F^{-1}, y) = \int_0^1 2\Lambda_\kappa(F^{-1}(\kappa), y)d\kappa,$$

where $\Lambda_\kappa(q, y) = (\kappa - \mathbb{1}_{\{y<q\}})(y - q)$ is the pinball loss for a specific quantile level $\kappa$. The CRPS measures the compatibility between the predicted inverse cumulative distribution function (also known as the quantile function), $F^{-1}$, and the observation, $y$. In practice, the quantile function is not analytically available and the CRPS is approximated using discrete quantile levels derived from samples. We used the implementation in GluonTS [2], which defaults to nine quantile levels, $\{0.1, 0.2, 0.3, 0.4, 0.5, 0.6, 0.7, 0.8, 0.9\}$, estimated using 100 sample forecasts.

**Linear Predictive Score (LPS).** The LPS is our proposed metric to evaluate the predictive quality of synthetic time series samples from a generative model. We define the LPS as the test CRPS of a linear ridge regression model trained on the synthetic samples. We trained the ridge regression model using 10,000 synthetic samples and computed the CRPS on the corresponding real test set. Note that the CRPS of a point forecaster, such as the linear model, is equal to the $0.5$-quantile loss (also known as the Normalized Deviation).

## B.3 Hyperparameters and Training Details

We trained TSDiff using the Adam optimizer for 1,000 epochs with a learning rate of 1e-3. Every epoch is comprised of 128 batches constructed with 64 sequences each. We used 3 residual layers with 64 channels in the backbone of TSDiff. We also added a skip connection from input to output for some datasets as we observed improvements in the validation performance. The number of timesteps in the diffusion process was set to $T = 100$ and we used a linear scheduler with $\beta_1 = 0.0001$ and $\beta_{100} = 0.1$. The diffusion timesteps were encoded into 128-dimensional positional embeddings [53]. Table 6 provides an overview of the hyperparameters.

Table 6: Hyperparameters of TSDiff.

| Hyperparameter | Value |
| --- | --- |
| Learning rate | 1e-3 |
| Optimizer | Adam |
| Batch size | 64 |
| Epochs | 1000 |
| Gradient clipping threshold | 0.5 |
| Residual layers | 3 |
| Residual channels | 64 |
| Time Emb. Dim. | 128 |
| Diffusion steps $T$ | 100 |
| $\beta$ scheduler | Linear |
| $\beta_1$ | 0.0001 |
| $\beta_{100}$ | 0.1 |

**Normalization.** The magnitudes of the time series can vary significantly, even within the same dataset. Training on raw values of such time series can lead to optimization instabilities. Therefore, we normalized the time series before training. Specifically, we used the mean scaler in GluonTS [2] which divides each time series individually by the mean of the absolute values of its observed context,

$$\mathbf{y}_{\text{obs}}^{\text{norm}} = \frac{\mathbf{y}_{\text{obs}}}{\text{mean}(\text{abs}(\mathbf{y}_{\text{obs}}))},$$

where $\text{abs}(\cdot)$ is the element-wise absolute function and $\text{mean}(\cdot)$ denotes the mean aggregator. After inference, i.e., after computing $\mathbf{y}_{\text{ta}}^{\text{norm}}$, we rescale the target back to the original magnitude,

$$\mathbf{y}_{\text{ta}} = \mathbf{y}_{\text{ta}}^{\text{norm}} \cdot \text{mean}(\text{abs}(\mathbf{y}_{\text{obs}})).$$

**Guidance Scale Selection.** The guidance scale in observation self-guidance (see Alg. 1) controls the strength of guidance and was selected on the basis of the validation performance. We tuned the scale from the set $\{2/32, 3/32, 4/32, 5/32\}$ for mean square guidance and observed that it performs well with the scale $4/32$ on every dataset. For quantile guidance, we tuned the scale from the set $\{1, 2, 4, 8\}$. Table 7 reports the best performing scale for quantile guidance on each dataset. The guidance scale also controls the alignment between the diffused time series and the observations. Fig. 5 shows that diffused time series from TSDiff-Q align closely with the observations.

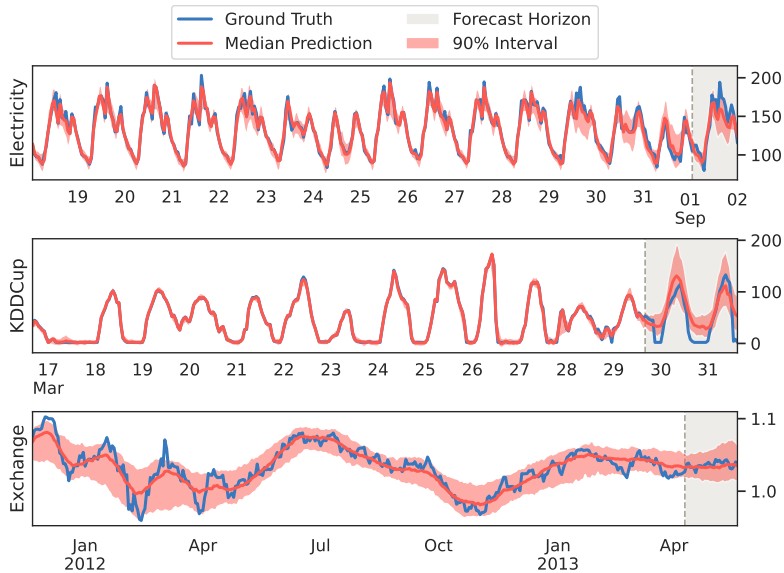

Figure 5: Predictions generated by TSDiff-Q for time series in Electricity, KDDCup, and Exchange.

Table 7: Guidance scales, $s$, for quantile self-guidance on each dataset.

| Dataset | Quantile Guidance Scale |
|---|---|
| Exchange | 8 |
| Solar | 8 |
| Electricity | 4 |
| Traffic | 4 |
| M4-Hourly | 2 |
| KDDCup | 1 |
| UberTLC-Hourly | 2 |
| Wikipedia | 2 |

**Forecasting with Missing Values.** As discussed in Sec. 4, we evaluated the forecasting performance of TSDiff under three missing value scenarios to demonstrate its flexibility. Specifically, we examined the ability of observation self-guidance to handle missing values during inference. For each dataset, we trained a single unconditional model for TSDiff-Q without any modifications to the training objective. During inference, we masked 50% of the timesteps in the context window during inference, e.g., 168 out of 336 timesteps on the Solar dataset. The selection of which timesteps to mask depended on the missing value scenario (see Fig. 6 for an illustration of the three scenarios). Observation self-guidance was then used to predict both the missing and the target values (i.e., the forecast). The CRPS was only computed for the forecast horizon. To ensure that TSDiff did not train on the timesteps that would be masked during inference, we excluded a window of length equal to $L$ (context length + prediction length) from the end of the training time series. Consequently, the M4-Hourly and Wikipedia datasets were excluded from the missing value experiments due to the resulting time series being too short.

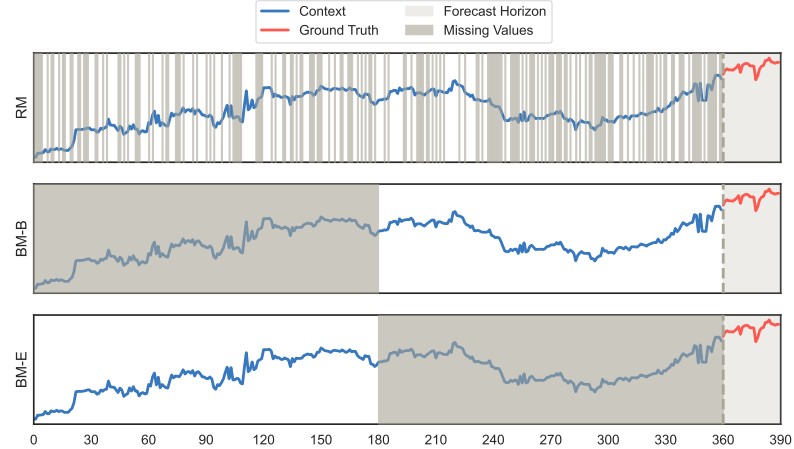

Figure 6: Illustration of the three missing value scenarios considered in our experiments.

On the other hand, the conditional models (TSDiff-Cond) were trained individually for each scenario, i.e., the missing values were masked and the model was optimized to predict them correctly.

**Computational Cost of Observation Self-Guidance.**    Table 8 reports the inference time of observation self-guidance against the conditional diffusion model (TSDiff-Cond) on the Exchange dataset while controlling for aspects such as batch size, number of samples and GPU. The additional overhead for observation self-guidance (i.e., TSDiff-MS and TSDiff-Q) comes from the computation of the gradient of a neural network (see the Eq. 10).

Table 8: Comparison of inference times of TSDiff-Cond and observation self-guidance on the Exchange dataset.

| Model | Inference Time (seconds) |
| --- | --- |
| TSDiff-Cond | 162.97 |
| TSDiff-MS | 201.08 |
| TSDiff-Q | 201.67 |

**Refinement Representative Step.**    As discussed in Sec. 3.2, we use a representative diffusion step to approximate the ELBO instead of randomly sampling multiple diffusion steps. The representative step corresponds to the diffusion step that best approximates the average loss. Fig. 7 shows the loss per diffusion step, the average loss and the representative step, $\tau$, for the eight datasets used in our experiments. We computed the loss per diffusion step on a randomly-sampled batch of 1024 datapoints, which took ≈13 seconds per dataset on a single Tesla T4 GPU.

**Refinement Iterations Selection.**    To strike a balance between convergence and computational efficiency, our refinement scheme necessitates careful selection of the number of refinement iterations. The relationship between the relative CRPS and the number of iterations is visualized in Fig. 8 for the Solar dataset. Notably, immediate improvement is observed after just one iteration for point forecasters like Seasonal Naive and Linear. Conversely, DeepAR and Transformer exhibit gradual enhancements. This analysis was conducted solely on the Solar dataset, determining that 20 refinement iterations offer a favorable balance between performance and computational requirements. As a result, we set the number of refinement iterations to 20 for all other datasets and base forecasters.

### B.4   Baselines

We compared against eleven baselines, both from statistical literature and deep learning-based models, for the forecasting experiments. In the following, we briefly describe the individual baselines.

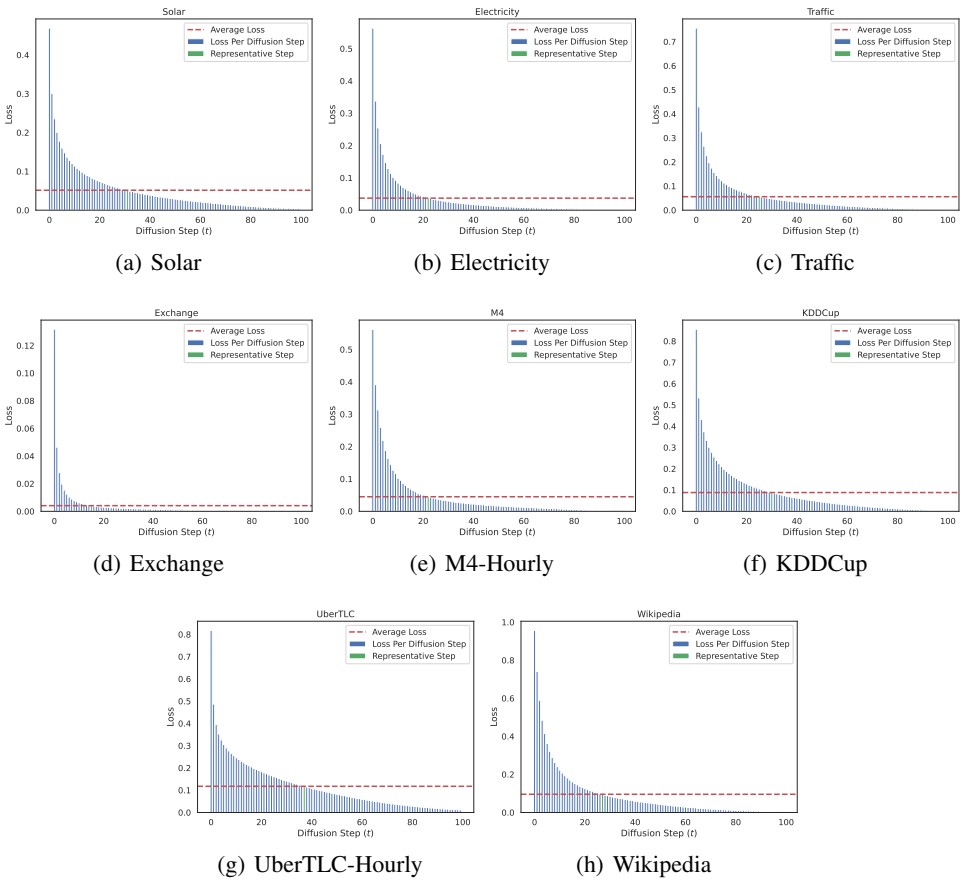

Figure 7: Loss per diffusion step, $t$, and representative diffusion step, $\tau$, for the eight datasets used in our experiments.

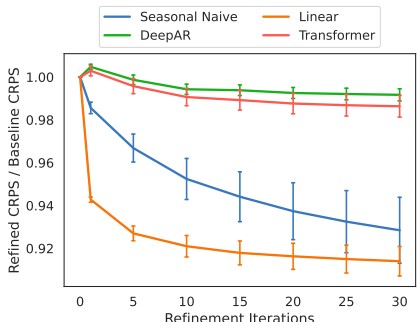

Figure 8: Variation of the relative CRPS with the number of refinement iterations on the Solar dataset.

- **Seasonal Naive** is a naive forecaster that returns the value from the last season as the prediction, e.g., the value from 24 hours (1 day) ago for time series with hourly frequency.
- **ARIMA** [21] is a popular statistical model for analyzing and forecasting time series data, combining autoregressive (AR), differencing (I), and moving average (MA) components to capture underlying patterns and fluctuations. We used the implementation available in the `forecast` package [22] for R.
- **ETS** [21] is a forecasting method that uses exponential smoothing to capture trend, seasonality, and error components in the time series. We used the implementation available in the `forecast` package [22] for R.

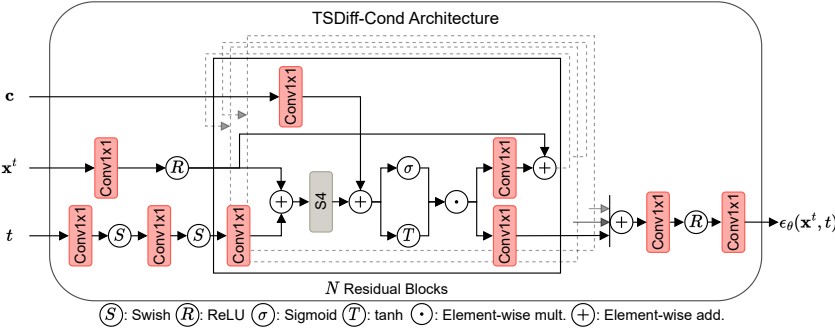

Figure 9: A schematic of the architecture of the conditional model, TSDiff-Cond. The key difference from TSDiff is the incorporation of the conditioning input, **c**, through Conv1x1 layers.

- **Linear** is a linear ridge regression model across time, i.e., it takes the past context length timesteps as inputs and outputs the next prediction length timesteps. We used the implementation available in `scikit-learn` [40] with the regularization strength of 1 (its default value in `scikit-learn`). The model was fit using 10,000 randomly sampled sequences with the same context length for each dataset as for TSDiff.

- **DeepAR** [46] is an RNN-based autoregressive model that is conditioned on the history of the time series through lags and other relevant features, e.g., time features such as hour-of-day and day-of-week. The model autoregressively outputs the parameters of the next distribution (e.g., Student's-t and Gaussian) and is trained to maximize the (conditional) log-likelihood. We used the implementation available in GluonTS [2] with the recommended hyperparameters.

- **MQ-CNN** utilizes a convolutional neural network (CNN) architecture to capture patterns and dependencies within the time series data. The model is trained using the quantile loss to directly generate multi-horizon forecasts. We used the implementation available in GluonTS [2] with the recommended hyperparameters.

- **DeepState** [42] combines RNNs with linear dynamical systems (LDS). The RNN takes additional covariates (e.g., time features and item ID) as input and outputs the (diagonal) noise covariance matrices of the LDS. Other parameters of the LDS such as the transition and emission matrices are manually designed to model different time series components such as level, trend, and seasonality. The LDS and RNN are jointly trained via maximum likelihood estimation. We used the implementation available in GluonTS [2] with the recommended hyperparameters.

- **Transformer** is a sequence-to-sequence forecasting model based on the self-attention architecture [53]. The model takes time series lags and covariates as inputs and outputs the parameters of future distributions. We used the implementation available in GluonTS [2] with the recommended hyperparameters.

- **TFT** [32] is another sequence-to-sequence forecasting model based on the self-attention architecture that includes a variable selection mechanism to minimize the contributions of irrelevant input features. The model is trained to minimize the quantile loss at the selected quantiles. We used the implementation available in GluonTS [2] with the recommended hyperparameters.

- **CSDI** [52] is a conditional diffusion model for multivariate time series imputation and forecasting. Similar to TSDiff, CSDI is based on the DiffWave architecture [28], but utilizes transformer layers as fundamental building blocks. The original CSDI architecture was presented in the context of multivariate time series comprising temporal and feature transformer layers. We used the original implementation[13] in the univariate setting with the recommended hyperparameters and training setup in our experiments.

- **TSDiff-Cond** is a conditional version of TSDiff. It integrates the observed context together with an observation mask using a Conv1x1 layer followed by an addition after the S4 layer

---

[13]CSDI implementation: https://github.com/ermongroup/CSDI

in each residual block as shown in Fig 9. TSDiff-Cond's architecture resembles that of SSSD [1] with three key differences:

- TSDiff-Cond uses S4 layers in the residual blocks before addition with the conditioning vector whereas SSSD uses them both before and after addition.
- TSDiff-Cond is a univariate model where feature dimensions indicate lagged time series values whereas SSSD is a multivariate model with feature dimensions indicating the different features in the multidimensional time series.
- TSDiff-Cond computes the loss on the entire time series, unlike SSSD which only generates the unobserved timesteps. This led to better performance of the conditional model in our experiments.

We used the same hyperparameters (e.g., context lengths and lags) for TSDiff-Cond as for TSDiff (i.e., the unconditional model) to isolate the effect of conditional training.

For the train on synthetic-test on real experiments, we compared against two popular time series generative models.

- **TimeVAE** is based on variational autoencoders and includes network components specific to time series such as trend and seasonality blocks. We used the original implementation[14] and recommended hyperparameters in our experiments.
- **TimeGAN** employs an autoencoder to train a generative adversarial network in the latent space. The network components are trained using a combination of supervised, unsupervised, and discriminative loss functions. We used the original implementation[15] and recommended hyperparameters in our experiments.

## C  Synthetic Samples

Figs. 10 to 17 illustrate synthetic time series created by TimeVAE, TimeGAN, and TSDiff in comparison to real time series. The generated samples from TSDiff closely resemble the real samples across all datasets, exhibiting higher quality compared to the baselines. For instance, in the case of the solar dataset (Fig. 10), TSDiff accurately captures periods of zero solar energy production during nighttime and generates a wide range of peaks within individual time series. In contrast, samples from TimeVAE and TimeGAN appear more homogeneous and fail to capture the zero values precisely. TimeGAN also experiences mode collapse issues in certain datasets like Electricity, M4-Hourly, and UberTLC-Hourly, while TSDiff consistently generates diverse and realistic samples.

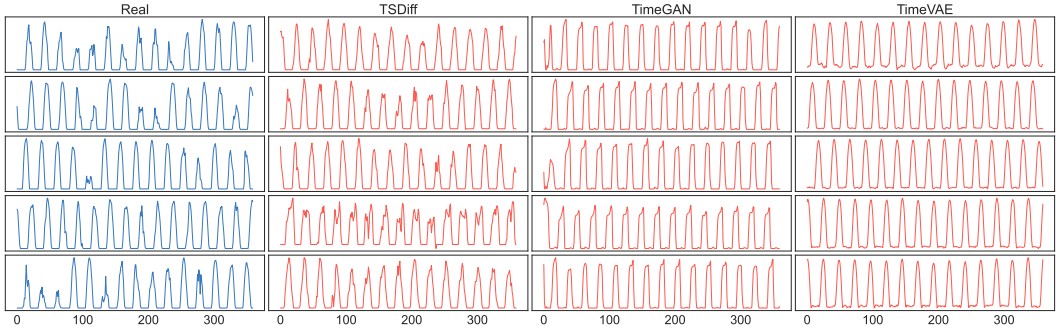

Figure 10: Real samples and synthetic samples generated by TSDiff, TimeGAN, and TimeVAE for the Solar dataset.

---

[14]TimeVAE implementation: https://github.com/abudesai/timeVAE
[15]TimeGAN implementation: https://github.com/jsyoon0823/TimeGAN

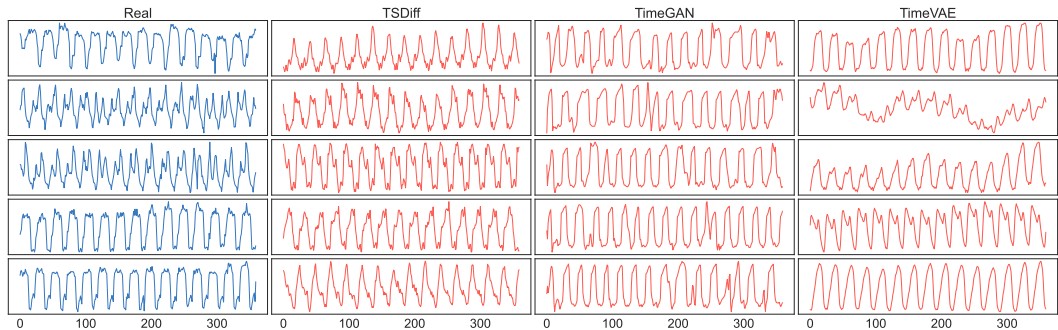

Figure 11: Real samples and synthetic samples generated by TSDiff, TimeGAN, and TimeVAE for the Electricity dataset.

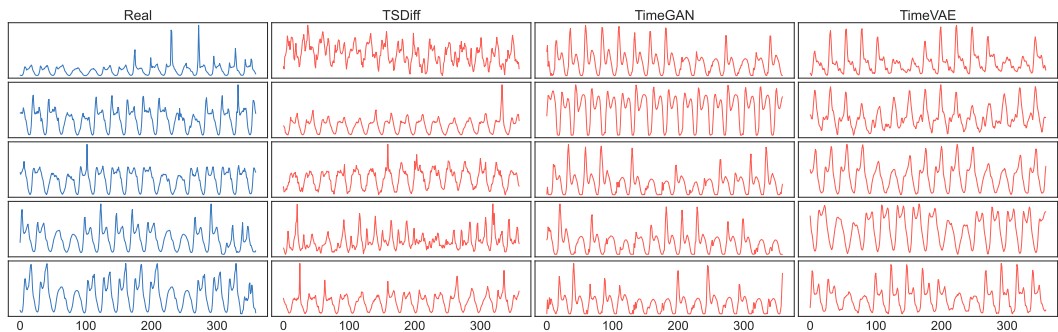

Figure 12: Real samples and synthetic samples generated by TSDiff, TimeGAN, and TimeVAE for the Traffic dataset.

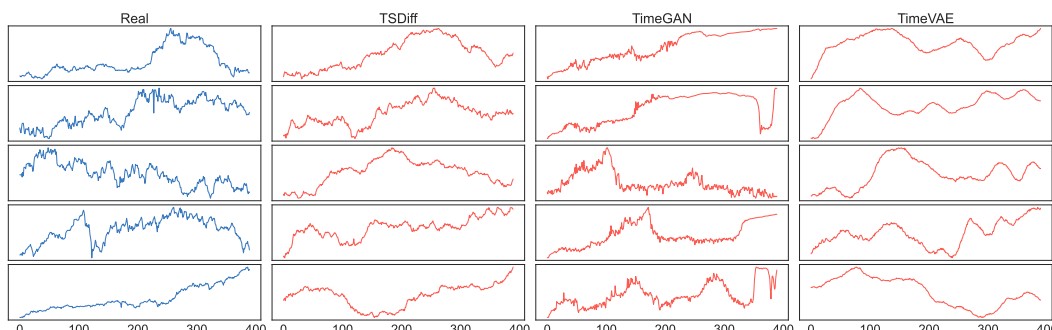

Figure 13: Real samples and synthetic samples generated by TSDiff, TimeGAN, and TimeVAE for the Exchange dataset.

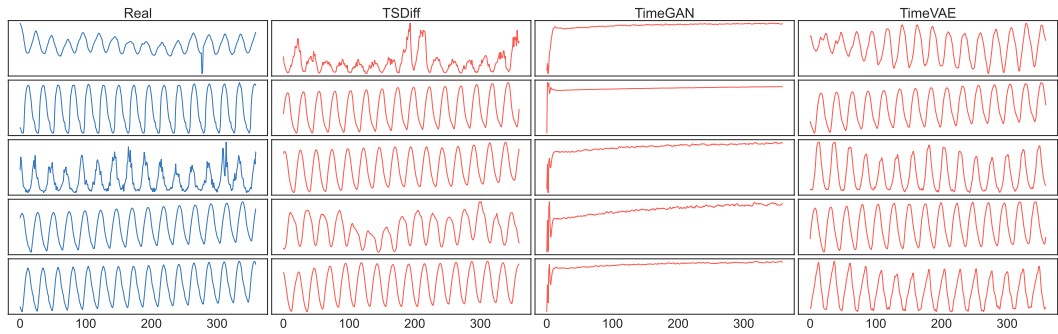

Figure 14: Real samples and synthetic samples generated by TSDiff, TimeGAN, and TimeVAE for the M4-Hourly dataset.

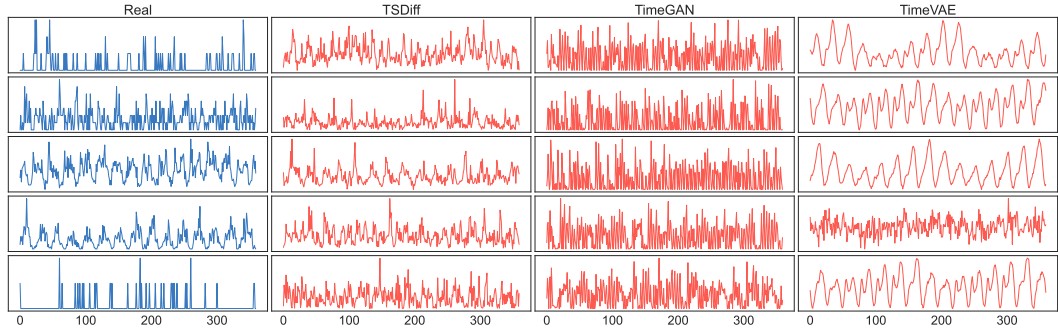

Figure 15: Real samples and synthetic samples generated by TSDiff, TimeGAN, and TimeVAE for the UberTLC-Hourly dataset.

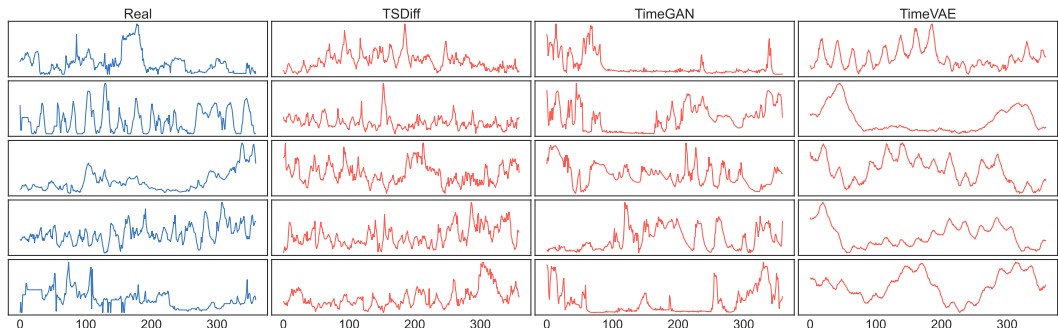

Figure 16: Real samples and synthetic samples generated by TSDiff, TimeGAN, and TimeVAE for the KDDCup dataset.

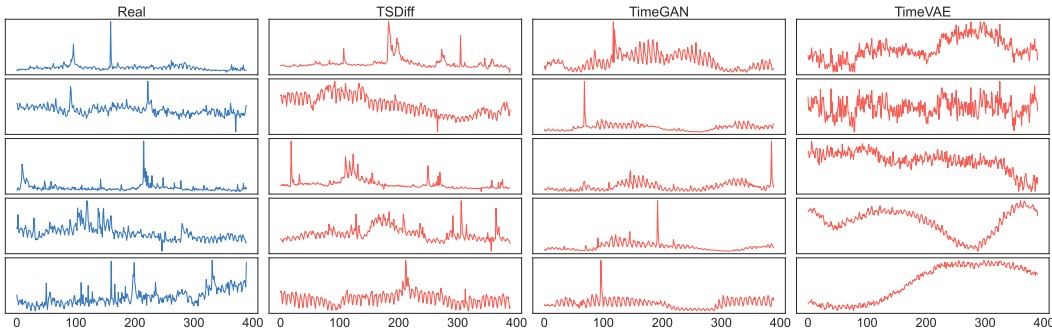

Figure 17: Real samples and synthetic samples generated by TSDiff, TimeGAN, and TimeVAE for the Wikipedia dataset.

