# OpenReview forum: "Predict, Refine, Synthesize: Self-Guiding Diffusion Models for Probabilistic Time Series Forecasting"
_NeurIPS.cc/2023/Conference — NeurIPS 2023 poster_

### Official Review · Reviewer_SUq5 · 2023-07-05

**Soundness:** 2 fair
**Presentation:** 2 fair
**Contribution:** 2 fair
**Rating:** 4
**Confidence:** 3

**Summary:**

The paper focuses on generative diffusion modeling when some of the samples are known, which is advocated as the right setup for regression. Indeed, in a time-series context, those known samples may lie in the past (prediction), or provide boundary conditions (interpolation).
In this context, the classical diffusion equations can be worked out to yield a generation that is compatible with the known samples to some controllable extent. This is the key contribution for this paper, and I must say that traming regression as a constrained diffusion is an interesting  idea.

From this starting point, the authors try to derive two variants: one where the model estimate is used as a prior, and one where some arbitrary other estimate is used.

A set of forecasting experiments conclude the paper

**Strengths:**

- the paper is very readable and provides some background and references. It is quite clear
- Framing regression as a constrained diffusion is definitely interesting and has some merits.
- I would also say that I'm sure that the proposed method yields interesting results, compared to some other methods

**Weaknesses:**

Notwithstanding its qualities, I have several concerns with the paper that prevent me from giving it a high score.
- I would say that the attempt at formulating the method in a Bayesian way is actually a bit awkward in my opinion, because it does not bring much more food for thought than just plainly presenting the guidance as a regularization over the (enegry-based) generating process. Indeed, the prior distribution that is picked in equation (5) just reads like picking some L2 regularization to me, and the quantile regularization that is advocated as working very well in practice just doesn't lend itself very well to a convenient Bayesian treatment. Actually, section 3.2 precisely drops this Bayesian vision to adopt an optimization-based point of view, which seems more natural to me.

- I feel that there is actually a lot of redundancy between 3.1 and 3.2. As I see it, the two "variants" are actually the same, with the only difference that for self-guidance, the estimate is derived from the model while it is assumed arbitrary in 3.2.

**Questions:**

- L85: consists of decomposing
- L119: "see Eq 1": It is not clear to me why you provide this reference to eq (1)
- eq (5), again, the choice of a unit variance for the likelihood is weird to me. I would say this is only picked this way to motivate some L2 regularization term in a "reverse mathemating" sense
- I don't understand how you get \epsilon in equation (10) and further during inference time.
- L249-250: when you write that the model does not require task-specific training, could you actually explicitly mention *what* is the model trained on ? I thought that there would be a separate training for each task ?


**Limitations:**

- will you provide some implementation ?

---

> ### Author Rebuttal · Authors · 2023-08-09
>
> We thank the reviewer for their review and positive comments on the writing and the merits of our ideas. In the following, we respond to specific questions and concerns raised by the reviewer.
>
> **Comment:** On the Bayesian view of self-guidance and Eq. 5.
> **Response:** We introduced self-guidance using the Bayes' rule to retain the probabilistic perspective of the guidance term, and to keep the story and notation consistent with the original work on classifier guidance [1]. In our opinion, the Bayesian view is a cleaner and less ad-hoc way to set up the method. We later clarify in the paper that the selected distributions result in the MSE and quantile losses (see L138 and L148). We dropped the Bayesian view in section 3.2 because the _data-space_ refinement scheme makes it confusing to set it up.
>
> **Comment:** Redundancy between 3.1 (self-guidance) and 3.2 (refinement). "... the two "variants" are actually the same, with the only difference that for self-guidance, the estimate is derived from the model while it is assumed arbitrary in 3.2."
> **Response:** We believe that this is not an accurate characterization of the differences between our self-guidance and refinement schemes. The diffusion model estimate is used in *both* schemes. What differs is the interpretation and the sampling procedure.
> - **Self-guidance** modifies the reverse diffusion process with a self-conditioning score function. Ancestral sampling is performed starting from noise at step $T$ all the way to the observation space at step $0$ by iteratively denoising and guiding the sample.
> - **Refinement** interprets the implicit density of the diffusion model as a regularized energy landscape. Rather than starting from noise, it starts from the prediction of a base model and uses Langevin dynamics to sample _directly in the data space_.
>
> The self-guidance and refinement schemes can also be viewed as the predictor and corrector schemes used in some score-based diffusion models [2].
>
>
> **Comment:** Why refer to Eq. 1 in L119?
> **Response:** The neural network $\epsilon_\theta$ is trained on noisy time series $\mathbf{x}^t\in\mathbb{R}^{L\times C}$. These noisy time series are obtained using Eq. 1, hence the reference to it.
>
> **Comment:** How to get $\epsilon$ in Eq. 10?
> **Response:** $\epsilon$ is drawn from the standard Gaussian distribution, wheras $\epsilon_\theta$ is the denoising neural network (see Fig. 2). $\mathbf{x}^t$ is the noisy transformation of $\mathbf{y}$ obtained using Eq. 1. As mentioned in L187, Eq. 10 is a simplification of the ELBO and serves as an approximation of $\log p_\theta(\mathbf{y})$.
>
> **Comment:** "What is the model trained on? I thought that there would be a separate training for each task?"
> **Response:** The model is trained as an unconditional generative model for each dataset. This means that the model is trained to denoise the complete sequence, $\mathbf{x}^t$. In the *forecasting with missing values* experiment, we trained one model per dataset and used the same model for inference on the different missing value tasks (e.g., random missing, blackout missing). This is in contrast to the conditional models that are trained for specific missing value tasks. Please note that the term "task" in our context does not refer to the one used in meta-learning and foundation models literature, as clarified in footnote 1.
>
> **Comment:** Will the implementation be available?
> **Response:** The implementation has been provided as supplementary material and will be made publicly available at a later stage.
>
> We thank the reviewer again for their feedback and hope that we have satisfactorily addressed their concerns regarding the Bayesian view and the difference between self-guidance and refinement. If so, _we hope that the reviewer would consider raising their score_.
>
> [1] **Dhariwal, Prafulla, and Alexander Nichol**. "Diffusion models beat gans on image synthesis." Advances in neural information processing systems 34 (2021): 8780-8794.
>
> [2] **Song, Yang, Jascha Sohl-Dickstein, Diederik P. Kingma, Abhishek Kumar, Stefano Ermon, and Ben Poole**. "Score-based generative modeling through stochastic differential equations." arXiv preprint arXiv:2011.13456 (2020).

---

### Official Review · Reviewer_8t5R · 2023-07-05

**Soundness:** 3 good
**Presentation:** 3 good
**Contribution:** 3 good
**Rating:** 5
**Confidence:** 4

**Summary:**

This paper proposes TSDiff, which is an unconditional diffusion model for time series generation. Besides, the authors propose self-guidance and prediction refinement. The empirical results showcase the superiority of TSDiff over existing baselines on forecasting, refinement, and generating synthetic samples.

**Strengths:**

1. The paper is joyful to read and the methodology is clearly described.
2. The empirical results are more than enough. The authors conduct extensive experiments on various of tasks and datasets, making the proposed methodology very convincing.
3. It is good to see that the authors proposed a new metric to make more reasonable comparison with other baselines in Section 5.3.

**Weaknesses:**

1. If y_obs is a subset of timesteps representing the observed timesteps, then is the conditional generation similar to an inpainting problem in 2D images? Then important baselines in solving inverse problems with pre-trained diffusion models, e.g., DDRM [1], should be included, or at least be discussed thoroughly.

2. There are several equations that are not labeled with numbers. It would be the best if the authors can label them. I have several questions regarding the equation after Eq. (10):

[1] Is tau hard to solve? It looks like solving tau is computationally expensive. It would help if the authors can provide some empirical results. Actually I did not quite understand "tau can be computed efficiently by tracking the running mean for the losses at all diffusion steps during training".

[2] How does the optimum, tau, vary with different data point? In this equation, the expectation does not take x_t (or y) into account, does this mean we have to solve for tau for every y? Or do we jointly solve a tau using all the data points in the training set?

**Questions:**

See Weaknesses.

**Limitations:**

The limitations are well discussed in the paper.

---

> ### Author Rebuttal · Authors · 2023-08-09
>
> We thank the reviewer for their review and positive comments on our method and writeup. In the following, we respond to specific concerns and questions raised.
>
> **Comment:** "...is the conditional generation similar to an inpainting..." and discussion of image inpainting models.
> **Response:** At a high level, there are indeed similarities between the time series forecasting/imputation problems we address and the inpainting problem in 2D images. Both involve generating missing or unobserved data conditioned on the observed data. We acknowledge the similarities and cite prior work on inpainting in our related work discussion (see L198). However, there are also key differences. In time series problems, the temporal dynamics and the sequential nature of the data introduce complexities and challenges that are not present in the static 2D inpainting problem. While DDRM provides a powerful framework for linear inverse problems, directly applying it to our problem might not be straightforward due to the aforementioned complexities. That said, we agree that it is an important related work and we will include a discussion on DDRM in the revised manuscript.
>
>
> **Comment:** Missing labels for equations.
> **Response:** We only added labels for equations that we refer to in the text. We will add labels to all equations in the revision.
>
> **Comment:** Computational complexity of the representative diffusion step, $\tau$.
> **Response:** $\tau$ is not computationally expensive to solve. There are two ways to obtain $\tau$:
> - **After training**: Compute the average loss over random datapoints for each diffusion step $t$. Then, select the $t$ closest to the average loss over all diffusion steps as the representative step, $\tau$. This can be done by the following line of code where `losses` is an array of the loss for each diffusion step.
> ```python
> tau = ((losses - losses.mean()) ** 2).argmin()
> ```
> As a result, we end up with one $\tau$ for the entire dataset.
>
> In our experiments, we computed the loss per diffusion step with a randomly sampled batch of 1024 datapoints. The computation of $\tau$ took around 13 seconds per dataset on a single Tesla T4 GPU. We will add these details to the revision.
> - **During training**: Rather than computing the losses for each $t$ after the training, we can keep a running average loss for each $t$ and compute the representative timestep using these losses, as above. This is possible because the loss we use to obtain $\tau$ is the same as the training loss. We will clarify the text in the revision.
>
> **Comment:** "...the expectation does not take x_t (or y) into account..." How does the optimum ($\tau$) vary with different data points?
> **Response:** Thank you for spotting this typographical error. There should be an expectation over the datapoints as well. We will correct this in the revision.
>
> As mentioned above, a single diffusion step $\tau$ is computed per dataset and it does not vary with data points. Fig. 2 in the additional rebuttal PDF shows the loss per diffusion step, the average loss and the resulting representative timestep for the 8 datasets used in our experiments.
>
> We thank the reviewer again for their review and suggestions. We hope that we have addressed the reviewer's concerns regarding the computational complexity of the representative diffusion step. If so, _we kindly ask the reviewer to consider raising their score_.

---

> > ### Comment · Reviewer_8t5R · 2023-08-14
> > **Thank you**
> >
> > Thank you for the additional results and clarifications. I will keep my rating and I tend to accept this paper.

---

> > > ### Author Response · Authors · 2023-08-14
> > >
> > > Thank you for your response. If you have any further questions, we will be happy to answer them.

---

### Official Review · Reviewer_D1QU · 2023-07-06

**Soundness:** 3 good
**Presentation:** 2 fair
**Contribution:** 3 good
**Rating:** 7
**Confidence:** 4

**Summary:**

This paper describes a diffusion model for time series problems. Contrary to the popular approach of using a conditional diffusion model, the authors proposed to use the unconditional diffusion model, which is supplemented by a self-guidance mechanism. The authors also proposed a prediction refinement algorithm to improve the prediction of any time series model by using the trained diffusion model.

**Strengths:**

Training a conditional diffusion model is challenging due to the high dimensionality of the conditional problem. The authors proposed an interesting method of bypassing training a conditional diffusion model. The proposed methodology is simple and sound.

**Weaknesses:**

The paper does not elaborate the details of their model, which makes it difficult to fully understand the methodology. For example, it does not clearly show the dimensions of input and output variables, how the missing inputs are treated, and so on.  There are also a few comments in Questions sections.

**Questions:**

1. In equation (6), as $t$ gets larger, the denominator $\sqrt{\overline{\alpha}_t} \rightarrow 0$, which is a natural consequence of the diffusion process. How do you handle this singularity issue? In practice, it will not be singular, but potentially cause a numerical instability.

2. The target distribution, $p(y_{ta}|y_{obs})$, but it actually should be $p(y_{ta}|y_{obs}) = p(y_{unobs}|y_{obs})p(y_{obs}|y_{obs})$ and $p(y_{obs}|y_{obs}) = \delta(y-y_{obs})$. So, upon the guidance denoising, we expect the observed $y$ should be converged to $y_{obs}$. I wonder if the model guarantees this.

3. Provide the proof that equation (9) converges to the maximum likelihood solution when $\gamma = 0$. Also, please provide a proof that (9) converges to $p(y)$.

4. Carefully reading it, I can understand how the guided diffusion, eqn (5), is computed for missing data and prediction. But, it will be useful if the authors can elaborate it to improve the readability.

5. It is claimed that the model can be used for imputation. Is the diffusion model trained with a missing data? Or, does it require a full data?

*I am willing to upgrade my score once these questions are addressed.

**Limitations:**

One of the questions of time series diffusion model is whether it memories the pattern or learns the dynamics. Based on the learning objectives, it is more likely to learn the patterns, which may limit the capability of diffusion-based models for the time series problems of chaotic dynamics and a stochastically forced system. Also, as the authors suggested, the computational cost is certainly of concern for a scalability. But these are beyond the scope of the study.

---

> ### Author Rebuttal · Authors · 2023-08-09
>
> We are grateful to the reviewer for their comprehensive review and appreciation of our work. Our response to specific questions raised by the reviewer follows.
>
> **Comment:** On model details (input and output dimensions, how missing values are treated, etc.).
> **Response:**
> - The denoising network takes an $L \times C$ dimensional input, as mentioned in L119, where $L$ is the sequence length and $C-1$ is the number of lags. As is typical in unconditional diffusion models, the output dimension is same as the input dimension.
> - During inference, the missing dimensions are masked when the self-guidance term is computed. Details on missing value experiments can be found in Appendix B3.
>
> We will further elaborate the model details in the revision. Specific implementation details can also be found in our code which has been released as part of the supplementary material.
>
> **Comment:** How is the singularity of $\sqrt{\bar{\alpha}_t}$ handled in Eq. 6?
> **Response:** We used a linear beta scheduler with $\beta_1=0.0001$ and $\beta_T=0.1$ with $T=100$ in our experiments. This results in $\sqrt{\bar{\alpha}_T}=0.075$ which did not lead to numerical instabilities. However, in our earlier experiments using the cosine scheduler [1], we observed unstable behavior with $\sqrt{\bar{\alpha}_T}=0.0002$. Based on this experience, numerical instability can be avoided by appropriately choosing $\beta_1$, $\beta_T$, $T$ and the beta scheduler.
>
>
> **Comment:** Does the observed section of $y$ converge to $y_\textrm{obs}$?
> **Response:** Self-guidance enforces a _soft-constraint_ on the observed timesteps. Therefore, the diffused values are not guaranteed to be exactly equal to the observations. The alignment between the predictions and observations for the observed timesteps can be controlled by the scale parameter $s$ (see Eq. 3). In practice, the diffused values for the observed timesteps are close to the observations, as shown in Fig. 1 in the rebuttal PDF.
>
> **Comment:** Proof of convergence of maximum likelihood and Langevin Monte Carlo.
> **Response:**
> - Note that Eq. 9 with $\gamma=0$ corresponds to gradient descent which is not guaranteed to converge to the global optimum for general (non-convex) problems. It may converge to a different local optima, depending on the intialization. This behavior is apparent in our results in Table 3 where the scores vary for different base forecasters. We will clarify this in the revision to avoid confusion.
> - For an SDE of the form $dX_t=-\nabla E(X_t)dt + \sqrt{2\gamma}dB_t$, the invariant distribution is the Gibbs-Boltzmann distribution $p(x) \propto \exp\left(-\frac{E(x)}{\gamma}\right)$, where $E$ is the energy function (equal to $-\log p_\theta(\mathbf{y}) + \lambda\mathcal{R}(\mathbf{y},\tilde{\mathbf{y}})$ in our case) and $B_t$ is a Brownian motion. This result can be derived using the Fokker-Plank Equation associated with the SDE (see Ch. 4 in [2]). The discretization of this SDE used in Eq. 9, also known as the _unadjusted Langevin algorithm (ULA)_, converges to the invariant distribution given certain regularity conditions (e.g., differentiability and Lipschitz gradients) on $E$ (see [3] for the analysis of ULA). Note that the invariant distribution in our case is not the true underlying distribution $p(\mathbf{y})$, which is unknown. Instead, we designed the energy function such that low energy is assigned to samples that are likely under the diffusion model $p_\theta(\mathbf{y})$ and also close to $\tilde{\mathbf{y}} = \mathrm{combine}(\mathbf{y}\_{\mathrm{obs}},g(\mathbf{y}\_{\mathrm{obs}}))$, ensured by the first and the second terms in the energy function, respectively. We will include this discussion in the revision.
>
> **Comment:** Is the model trained on missing or complete sequences?
> **Response:** The model was trained on complete sequences for the missing values experiments. The goal of this experiment was to evaluate the model's ability to handle/impute missing values during inference without any knowledge of the missing value patterns during training. We will clarify this in the revision. Further details on the missing values experiments (e.g., data splitting) can be found in Appendix B3.
>
> Note that while the model has been trained on complete sequences in our experiments, it is not a requirement for training TSDiff. Missing values during training can be easily handled by the S4 layers [4] used in our model by appropriately masking the missing timesteps.
>
> **Comment:** Does the model learn the dynamics or the patterns?
> **Response**: Additional investigation is needed to determine if the model learns the dynamics or patterns. Nevertheless, our results indicate that the model can make reliable predictions for typical time series forecasting datasets and tasks. Future research could focus on evaluating diffusion models' performance with chaotic/stochastically forced time series data, which could yield valuable insights.
>
> We thank the reviewer again for their insightful comments and valuable suggestions on improving the readability of our work. Based on our responses to the reviewer's questions on the specifics of our model, _we hope that the reviewer would consider raising their score_.
>
> [1] **Nichol, Alexander Quinn, and Prafulla Dhariwal**. "Improved denoising diffusion probabilistic models." In International Conference on Machine Learning, pp. 8162-8171. PMLR, 2021.
>
> [2] **Pavliotis, Grigorios A**. "Stochastic processes and applications." Springer-Verlag New York, 2016.
>
> [3] **Durmus, Alain, and Eric Moulines**. "Nonasymptotic convergence analysis for the unadjusted Langevin algorithm." (2017): 1551-1587.
>
> [4] **Gu, Albert, Karan Goel, and Christopher Ré**. "Efficiently modeling long sequences with structured state spaces." arXiv preprint arXiv:2111.00396 (2021).

---

> > ### Comment · Reviewer_D1QU · 2023-08-14
> >
> > While I believe that there is still room for improvement, the manuscript is interesting enough to guarantee acceptance. I changed my score.

---

> > > ### Author Response · Authors · 2023-08-14
> > >
> > > Thank you for your response and for increasing the score. We will be happy to hear any other suggestions that you may have regarding our manuscript.

---

### Official Review · Reviewer_PT4A · 2023-07-14

**Soundness:** 3 good
**Presentation:** 3 good
**Contribution:** 3 good
**Rating:** 7
**Confidence:** 4

**Summary:**

In this paper, the authors proposed an unconditional diffusion model and a self-guidance mechanism for time series data that can be used for conditioning diffusion model for downstream tasks, e.g. time series forecasting and imputation. The effectiveness of proposed model is evaluated from three aspects: prediction, refinement of prediction from other base forecaster, and time series generation. The results show that unconditional diffusion models can achieve comparable forecasting accuracy to conditional ones. The proposed model can be potential useful for other downstream time series analysis tasks other than studied ones.

**Strengths:**

1. The author proposed a new self-guidance mechanism to approximate conditional probability p(y_obs|x^t) which enables time series forecasting with unconditional diffusion model
2. Extensive experiments on multiple benchmarks demonstrate the effectiveness of the proposed unconditional diffusion model comparing to conditional ones.
3. The paper is well written and easy to follow.

**Weaknesses:**

1. Refining the prediction of a weak forecaster is a good way to evaluate the model, but it has no real application scenario.

**Questions:**

1. The authors described the details of how to do self-guidance without explanations of why this approximation can work in practice. It would be helpful to explain the high level intuition behind it.
2. The idea of self-guidance seems to be general, can it be used in other domain, e.g. image?
3. It is a good way to evaluate the model with refining the prediction. But this is not a practical senario. If one has the diffusion model, why not just use it for prediction? There are a lot of spaces used for this in section 4. It would be better for me to discuss other more interesting things.
4. The authors mentioned that the computational cost of the proposed model is high. It would be better to have such experiments comparing the forecasting speed with conditional diffusion models.
5. The authors evaluate the time series synthesis using train on synthetic-test on real setup. It would also be interesting to see train on synthetic and real - test on real setup. Can synthetic time series helpful for forecasting?
6. The experiments are conducted on univariate case, can the method be applied for multi-variate time series?
7. Can other time series analysis task get benefit from proposed models other than forecasting and imputation? For instance, time series classification, clustering, similarity search.

**Limitations:**

Yes, the author claimed the limitation of the paper, which is the computational cost.

---

> ### Author Rebuttal · Authors · 2023-08-09
>
> We thank the reviewer for their positive comments and constructive feedback. In the following, we respond to specific questions raised by the reviewer.
>
> **Comment:** Refinement has no real application scenario / Too much space dedicated to refinement.
> **Response:** The primary goal of the refinement experiments was to offer an alternative view on the diffusion model. Self-guidance modifies the reverse diffusion process whereas refinement utilizes the implicit probability density of the model. However, we respectfully disagree with the reviewer's comment that refinement lacks practical application scenarios.
>
> Refinement offers a complementary approach that caters to different computational constraints. While self-guidance generally yields superior results, it involves iterating through all $T$ diffusion steps, which can be computationally expensive. In contrast, refinement can be performed for fewer optimization steps, suitable to one's compute budget. Particularly, in the case of simple base forecasters, such as Seasonal Naive and Linear, even a single refinement step yields considerable improvements (see Fig. 5 in the Appendix). This highlights a valuable trade-off between runtime and prediction performance when compared to self-guidance. In industrial forecasting applications, it's not uncommon where one has access to a complex production forecasting system of black-box nature. Refinement provides a computationally efficient yet theoretically sound way to improve forecasts as a post-processor.
>
> **Comment**: What's the high-level intuition behind self-guidance? Why does the approximation work?
> **Response**: Guidance controls the reverse diffusion process via a conditioning term. Forecasting and imputation involve conditioning on the observed section of the time series. The main intuition behind "self"-guidance is that a model designed for complete sequences should reasonably approximate partial sequences. The one-step denoising used in Eq. 6 serves as a cost-effective approximation of the model for the observed time series, providing the requisite conditioning term for guidance in the form of the score function.
>
> **Comment:** Can self-guidance be used in other domains?
> **Response:** Yes, the idea of self-guidance is general and could potentially be used for other domains and tasks, e.g., images and videos. We are optimistic that future research will adopt our self-guidance mechanism across diverse domains.
>
> **Comment:** Computational cost of inference compared to conditional diffusion models.
> **Response:** Thank you for your suggestion. We will add a table in the appendix comparing the inference costs of conditional vs. unconditional diffusion models on different datasets. The following table shows the inference time for the conditional and unconditional models on the **Exchange** dataset while controlling for everything (e.g., batch size, number of samples, GPU).
>
> | Model    | Inference Time (seconds) |
> | -------- | -------- |
> | TSDiff-Cond | 162.97     |
> | TSDiff-MS | 201.08     |
> | TSDiff-Q | 201.67     |
>
> The additional overhead for the unconditional model (i.e., TSDiff-MS and TSDiff-Q) comes from the computation of the gradient of a neural network during self-guidance (see the equation after L128).
>
> **Comment:** Can a _train on real and synthetic_ scenario improve forecasting performance?
> **Response:** Based on our _train on synthetic-test on real_ results, we are hopeful that data augmentation with synthetic samples would improve downstream forecasting performance. Given other aspects of the current work (self-guidance and refinement), the scope of our generative evaluation was limited to the _train on synthetic-test on real_ scenario. This could be explored in future work.
>
> **Comment:** Can the method be applied to multivariate time series?
> **Response:** Yes, the ideas presented in this work are not limited to univariate time series. We decided to focus on the univariate case as univariate time series constitute a significant portion of real-world problems. The following are two simple ways of extending the model to handle multivariate time series:
> - Train the model with the channel-independence assumption (popularized by PatchTST [1]), i.e., all feature dimensions of the multivariate time series are treated as independent univariate time series.
> - Modify the backbone denoising network to incorporate a feature embedder for the multivariate time series.
>
> We will add this discussion to the revision.
>
> **Comment:** Can the proposed models benefit other time series tasks such as classification, clustering and similarity search?
> **Response:** Yes, we expect the proposed models to benefit other tasks. For example, by augmenting a classifier with synthetic samples or utilizing the imputation capabilities to clean datasets. Furthermore, future work could investigate the possibility of anomaly detection using the implicit density learned by the model.
>
> We thank the reviewer again for their comments and suggestions. We hope that we have satisfactorily answered the reviewer's questions and clarified sufficiently on the reviewer's main concern regarding the utility of refinement. If so, _we request the reviewer to consider raising their score_.
>
> [1] **Nie, Yuqi, Nam H. Nguyen, Phanwadee Sinthong, and Jayant Kalagnanam**. "A time series is worth 64 words: Long-term forecasting with transformers." arXiv preprint arXiv:2211.14730 (2022).

---

> > ### Comment · Reviewer_PT4A · 2023-08-14
> >
> > Thanks to the authors for their feedback. Still I am not fully convinced that the refinement of baseline forecaster like seasonal naive has practical impacts regarding computational constraint. The proposed method TSDiff could also be used with limited iterations and outputs a less accurate prediction, right? Then the authors would need to answer how do this one compared to the refinement of baseline forecasters with similar cost. Otherwise, maybe the authors could consider to sell the refinement point as a speed up or trade-off of accuracy and speed for the proposed TSDiff, which would be more convincing for me.

---

> > > ### Author Response · Authors · 2023-08-14
> > >
> > > Thank you for your response.
> > >
> > > We agree that the possibility of introduction of new forecasting models into specific real-world systems brings forth a need to study the trade-off between resource expenditure (in terms of computation) and performance. Specifically, this involves comparing the use of TSDiff as a standalone forecaster (with a limited number of iterations) against refining an existing base forecaster using TSDiff. However, in many industrial forecasting scenarios, factors such as resource limitations (pertaining to human capital), the presence of legacy systems and the downstream decision making system that relies on them, and stringent testing requirements often hinder or delay the replacement of existing production forecasting models. In these instances, refinement presents a cost-effective solution that enhances forecast accuracy post hoc without modifying the core forecasting process — a change that could potentially be a lengthy procedure. Importantly, refinement is versatile; it is independent of the base forecaster’s type and requires access only to forecast samples.
> > >
> > > > The proposed method TSDiff could also be used with limited iterations and outputs a less accurate prediction, right? Then the authors would need to answer how do this one compared to the refinement of baseline forecasters with similar cost.
> > >
> > > Comparing the performance of refinement and guidance might not be fair due to their distinct motivations and the reliance of refinement’s performance on the quality of underlying base forecasts. Nonetheless, we agree with the reviewer that this is an interesting experiment. We conducted an initial investigation using the first two datasets from our paper (Electricity and Solar). The table below presents a comparison of refinement from a seasonal naive predictor (LMC-Q, ML-Q) and diffusion guidance (TSDiff-Q) under different computational budgets (1, 2, 5 and 10 iterations).
> > >
> > > |   Iterations | Model          |   Electricity      |   Solar      |
> > > |-------------:|:---------------|-------------------:|-------------:|
> > > |             | Seasonal Naive |              0.069 |        0.512 |
> > > |              |                |                    |              |
> > > |            1 | LMC-Q          |          **0.054** |        0.505 |
> > > |            1 | ML-Q           |          **0.054** |    **0.504** |
> > > |            1 | TSDiff-Q       |              0.759 |        1.038 |
> > > |              |                |                    |              |
> > > |            2 | LMC-Q          |         **0.054** |        0.501 |
> > > |            2 | ML-Q           |         **0.054** |    **0.499** |
> > > |            2 | TSDiff-Q       |              0.816 |        1.013 |
> > > |              |                |                    |              |
> > > |            5 | LMC-Q          |              0.054 |        0.494 |
> > > |            5 | ML-Q           |          **0.053** |        0.493 |
> > > |            5 | TSDiff-Q       |              0.088 |    **0.483** |
> > > |              |                |                    |              |
> > > |           10 | LMC-Q          |              0.054 |        0.486 |
> > > |           10 | ML-Q           |          **0.052** |        0.485 |
> > > |           10 | TSDiff-Q       |              0.073 |     **0.419** |
> > >
> > > We observe that TSDiff-Q’s performance is poor with a low number of diffusion steps (1, 2), but improves when using 5 or more steps. In contrast, refinement demonstrates significant early-stage enhancements over the base model, which then plateau as more iterations are performed.
> > >
> > > We utilized the uniform skipping method proposed in [1] for diffusion guidance with reduced diffusion steps. The performance trends indicate that TSDiff-Q’s effectiveness increases with more iterations, while refinement provides considerable benefits during initial iterations before converging to a value.
> > >
> > > We hope that our response has convinced the reviewer of the practical utility of refinement. We will be happy to answer any further questions.
> > >
> > > [1] **Song, Jiaming, Chenlin Meng, and Stefano Ermon**. “Denoising diffusion implicit models.” arXiv preprint arXiv:2010.02502 (2020).

---

> > > > ### Comment · Reviewer_PT4A · 2023-08-15
> > > >
> > > > Thank you for the new experiments. Very interesting. I see your point about potential usage of refinement in production environment.

---

### Author Rebuttal · Authors · 2023-08-09

We thank the reviewers for their insightful reviews and their constructive feedback to improve the quality of our paper. We are pleased to note that the reviewers appreciate:

- the **technical significance of our self-guidance approach** ("an interesting method of bypassing training a conditional diffusion model", "constrained diffusion is definitely interesting and has some merits", rated *good* on **Soundness** and **Contribution** by 3/4 reviewers);
- the **clarity of our manuscript** ("paper is joyful to read and the methodology is clearly described", "the paper is very readable and provides some background and references. It is quite clear", "paper is well written and easy to follow");
- and, the **thoroughness of our empirical evaluation** ("Extensive experiments on multiple benchmarks demonstrate the effectiveness of the proposed unconditional diffusion model...", "The empirical results are more than enough ... making the proposed methodology very convincing", "I'm sure that the proposed method yields interesting results, compared to some other methods").

Detailed responses to individual reviewers are available under each review. In this general response, we seek to reemphasize the key contributions of our work.

- We present a fresh perspective on time series forecasting and imputation via *unconditional* diffusion modeling, in contrast to prior work that focuses on conditional models. To this end, we propose two novel inference schemes to utilize unconditional diffusion models for conditional tasks during inference.
    - **Observation self-guidance** conditions reverse diffusion for arbitrary forecasting tasks via a guidance term derived from the model's own estimate of the observed time series. The idea behind self-guidance is general and may be applicable to other domains.
    - **Refinement** uses the implicit density learned by the diffusion model to improve forecasts from base forecasters by sampling from an energy-based model.
- We show, through extensive benchmarks, that the proposed self-guidance approach is competitive against task-specific conditional models across datasets and forecasting scenarios. Furthermore, our refinement scheme is able to improve forecasts from base forecasters (especially the simple ones such as Linear and Seasonal Naive).
- We demonstrate empirically that downstream models trained solely on the synthetic samples from the diffusion models generate forecasts of high quality, outperforming existing time series generative models based on VAE and GAN frameworks. This opens up potential research directions analyzing the utility of the synthetic samples for downstream tasks either theoretically or empirically.
- We propose a metric to evaluate the quality of synthetic samples based on a simple Linear regression model which is not sensitive to architecture choices and random initializations, unlike existing predictive metrics.

---

### Author Response · Authors · 2023-08-17

Dear reviewers and AC(s),

Thank you once again for reviewing our manuscript and providing valuable feedback. We hope that our response has effectively addressed your inquiries and concerns. Should you have any additional questions, we will be happy to answer them.

Best regards,
Authors

---

### Decision · Program_Chairs · 2023-09-21

**Decision:**

Accept (poster)

**Comment:**

This paper proposes a diffusion model for time-series forecasting. Different from standard diffusion models, a self-guided mechanism is introduced to align the generation with observations. The model can perform prediction, refinement and synthesis.

Most reviewers think the method is interesting, They raised a number of concerns, which are well addressed in the rebuttal, and some reviewers raised their scores correspondingly. There is a one borderline reject, where the reviewer raised some questions such as writing issues. I believe these issues have been addressed in the rebuttal as well. Thus, I recommend acceptance.